# A degradative to secretory autophagy switch mediates mitochondria clearance in the absence of the mATG8-conjugation machinery

Hayden Weng Siong Tan [1,2], Guang Lu [1,3], Han Dong[1], Yik-Lam Cho [1], Auginia Natalia [4], Liming Wang[1,5], Charlene Chan[6], Dennis Kappei [6,7,8], Reshma Taneja [1,2], Shuo-Chien Ling [1], Huilin Shao [4], Shih-Yin Tsai [1], Wen-Xing Ding [9] & Han-Ming Shen [1,10]✉

PINK1-Parkin mediated mitophagy, a selective form of autophagy, represents one of the most important mechanisms in mitochondrial quality control (MQC) via the clearance of damaged mitochondria. Although it is well known that the conjugation of mammalian ATG8s (mATG8s) to phosphatidylethanolamine (PE) is a key step in autophagy, its role in mitophagy remains controversial. In this study, we clarify the role of the mATG8-conjugation system in mitophagy by generating knockouts of the mATG8-conjugation machinery. Unexpectedly, we show that mitochondria could still be cleared in the absence of the mATG8-conjugation system, in a process independent of lysosomal degradation. Instead, mitochondria are cleared via extracellular release through a secretory autophagy pathway, in a process we define as Autophagic Secretion of Mitochondria (ASM). Functionally, increased ASM promotes the activation of the innate immune cGAS-STING pathway in recipient cells. Overall, this study reveals ASM as a mechanism in MQC when the cellular mATG8-conjugation machinery is dysfunctional and highlights the critical role of mATG8 lipidation in suppressing inflammatory responses.

[1] Department of Physiology, Yong Loo Lin School of Medicine, National University of Singapore, Singapore, Singapore. [2] NUS Graduate School (Integrative Sciences and Engineering Programme), National University of Singapore, Singapore, Singapore. [3] Zhongshan School of Medicine, Sun Yat-sen University, Guangzhou, China. [4] Institute for Health Innovation & Technology, National University of Singapore, Singapore, Singapore. [5] School of Biomedical Sciences, Hunan University, Changsha, China. [6] Cancer Science Institute of Singapore, National University of Singapore, Singapore, Singapore. [7] Department of Biochemistry, Yong Loo Lin School of Medicine, National University of Singapore, Singapore, Singapore. [8] NUS Center for Cancer Research, Yong Loo Lin School of Medicine, National University of Singapore, Singapore, Singapore. [9] Department of Pharmacology, Toxicology and Therapeutics, The University of Kansas Medical Center, Kansas City, KS, USA. [10] Faculty of Health Sciences, University of Macau, Macau, China. ✉email: hmshen@um.edu.mo

Mitochondrial quality control (MQC) is critically important for maintaining mitochondrial homeostasis and normal physiological functions of the cell. In mammalian cells, MQC is achieved mainly via (i) ubiquitination and proteasomal degradation of mitochondrial proteins, (ii) autophagy, and (iii) mitochondrial-derived vesicles (MDVs)[1]. Among them, the selective degradation of mitochondria via the autophagosome–lysosome pathway (termed as mitophagy) is probably the most important pathway for the removal of damaged or dysfunctional mitochondria[2,3]. Multiple mechanisms have been established in the control of mitophagy and among them, mitophagy mediated by PTEN-induced kinase 1 (PINK1) and the ubiquitin E3 ligase Parkin is probably the most well studied[4–6]. During PINK1–Parkin-mediated mitophagy, PINK1 is stabilised and activated on the outer mitochondrial membrane (OMM) upon mitochondrial membrane depolarisation[7,8]. Subsequently, PINK1 targets both ubiquitin (Ub) and Parkin for phosphorylation and together with Parkin establishes a feed-forward loop, resulting in the mitochondrial accumulation of Parkin and coating of damaged mitochondria with phospho-Ub (S65) chains[9–12]. The p-Ub chains then serve to recruit autophagy receptors such as OPTN and NDP52 to damaged mitochondria[12–14], which subsequently mediate the formation of the autophagosome around the damaged mitochondria via the engagement of the autophagy machinery[15–17].

The autophagy machinery comprises of autophagy-related (ATG) proteins that are classified into a few functional groups. At the initiation stage, the ULK1 kinase complex (comprising of ULK1, FIP200, ATG13, and ATG101) is activated at the site of phagophore[15,18–20]. ATG9A is also recruited to support phagophore expansion[16,19]. Subsequently, the class III PI3K lipid kinase complex 1 (PI3KC3-C1; comprising of VPS34, VPS15, BECN1, ATG14) is recruited and activated to generate PtdIns(3)P (phosphatidylinositol 3-phosphate) on nascent autophagosome membranes[21–24], which serve to recruit downstream PtdIns(3)P effectors such as WIPI2 and DFCP1[23,24]. Downstream of this, the conjugation of mammalian ATG8s (mATG8s) to phosphatidylethanolamine (PE) on autophagosomal membranes represents a key feature in autophagy. In the mammalian cells, the mATG8 family consist of six orthologs, including the LC3 (LC3A, LC3B, and LC3C) and GABARAP (GABARAP, GABARAPL1, and GABARAPL2) subfamilies. The lipidation of mATG8s relies on the following two Ub-like conjugation systems[25–27]. In the ATG12 system, ATG12 is activated by the E1-like ATG7 before it is transferred to the E2-like ATG10, followed by its covalent conjugation to ATG5. The ATG12–ATG5 conjugate then interacts with ATG16L1 to form the ATG12–ATG5-ATG16L1 complex[28,29]. In the ATG8 system, ATG8 precursors are first cleaved by ATG4B in the cytosol[30,31] and ultimately conjugated to PE by the coordinated action of ATG7 (E1), ATG3 (E2), and the ATG12–ATG5-ATG16L1 complex (E3).

While ATG8 has been clearly demonstrated to be essential for autophagy in yeast[32], the requirement of mATG8s and its conjugation system in mammalian autophagy appears to be unclear. For instance, complete autophagosomes could be observed in various cell models that lacked mATG8s or components of its conjugation machinery[33–35]. Thus, it has been proposed that mATG8s are required for autophagosome membrane expansion, closure, autophagosome–lysosome fusion and degradation of the inner autophagosomal membrane, but not for the formation of autophagosomes per se. Additionally, a distinct autophagy pathway that is independent of ATG5 or ATG7 (termed as alternative autophagy) has been identified. In this pathway, autophagosome formation is initiated from the trans-Golgi network and relies on factors such as RAB9, WIPI3 and upstream autophagy factors ULK1 and BECN1, but not ATG9A[36,37].

Importantly, alternative autophagy has also been found to mediate the clearance of mitochondria from fetal reticulocytes during erythropoiesis, during iPSC reprogramming, and in cardiomyocytes under hypoxia or glucose deprivation conditions[38–40].

Although mitophagy appears to be the dominant mechanism for MQC, several other pathways have also been proposed. One such pathway involves the extrusion of mitochondria from cells into the extracellular milieu[41,42]. For instance, dysfunctional mitochondria have been found to be extruded from *C. elegans* neurons or mammalian cardiomyocytes via membrane-bound vesicles termed as exophers[43,44]. Additionally, damaged mitochondria were found to be released from multiple cell types in response to oxidative stress, mitochondrial depolarization or lipopolysaccharide treatment[45–47], as well as in disease status such as asthma and neurodegenerative disorders[48,49]. At present, the molecular mechanisms and functional implication of mitochondria extrusion have not been systematically examined. On the other hand, recent studies have demonstrated that secretory autophagy, a process involving autophagosome fusion with the plasma membrane instead of the degradative lysosome, results in the extracellular release of autophagosomal cargos[50,51]. To date, various protein cargoes (including IL1-β, ferritin, and Tau) have been identified to be secreted via secretory autophagy[51–54] and it remains to be tested whether secretory autophagy is implicated in clearance of mitochondria upon mitochondrial stress.

In this study, we aimed to examine the role of the mATG8-conjugation system in PINK1–Parkin-mediated mitophagy and explore alternative MQC pathways for mitochondria clearance. We first found that mitochondria clearance could still occur in the absence of key mATG8-conjugation machinery components, such as ATG7, ATG5, or ATG3, which are required for mATG8 lipidation. Unexpectedly, mATG8 lipidation-independent mitochondria clearance did not rely on lysosomal degradation. Instead, we discovered that this form of mitochondria clearance occurred via secretory autophagy, which we termed as Autophagic Secretion of Mitochondria (ASM). Functionally, ASM in the absence of the mATG8-conjugation system was found to produce a pro-inflammatory phenotype through the activation of the innate immune cGAS–STING pathway. Overall, we describe a MQC pathway whereby damaged mitochondria are routed towards autophagic secretion instead of autolysosomal degradation when the mATG8-conjugation machinery is defective. This study also highlights an important role of mATG8 lipidation in preventing the spurious activation of pro-inflammatory responses.

## Results

**The mATG8-conjugation system is not required for PINK1–Parkin-mediated mitochondria clearance.** To investigate the role of the mATG8-conjugation system in PINK1–Parkin-mediated mitophagy, we first established ATG7 knockouts using CRISPR/Cas9 genome editing in HeLa cells stably expressing mCherry-Parkin and mitoGFP (Supplementary Fig. 1a). Wild-type (WT) and ATG7-KO cells were then treated with a combination of antimycin A and oligomycin (A/O) to depolarise the mitochondria and activate the PINK1–Parkin pathway. To monitor the rate of mitochondria clearance, we examined the protein levels from various compartments of the mitochondria, including outer mitochondrial membrane (OMM) proteins, inner mitochondrial membrane (IMM) proteins and matrix proteins. As expected, treatment with A/O in WT cells resulted in time-dependent sequential degradation of mitochondrial proteins, from OMM (MFN1, TOM70), to IMM (COX-II, UQCRC2), and matrix proteins (citrate synthase [CS], mitoGFP), indicating a clearance of mitochondria (Fig. 1a, b). Additionally, the release of free GFP fragments from mitoGFP (akin

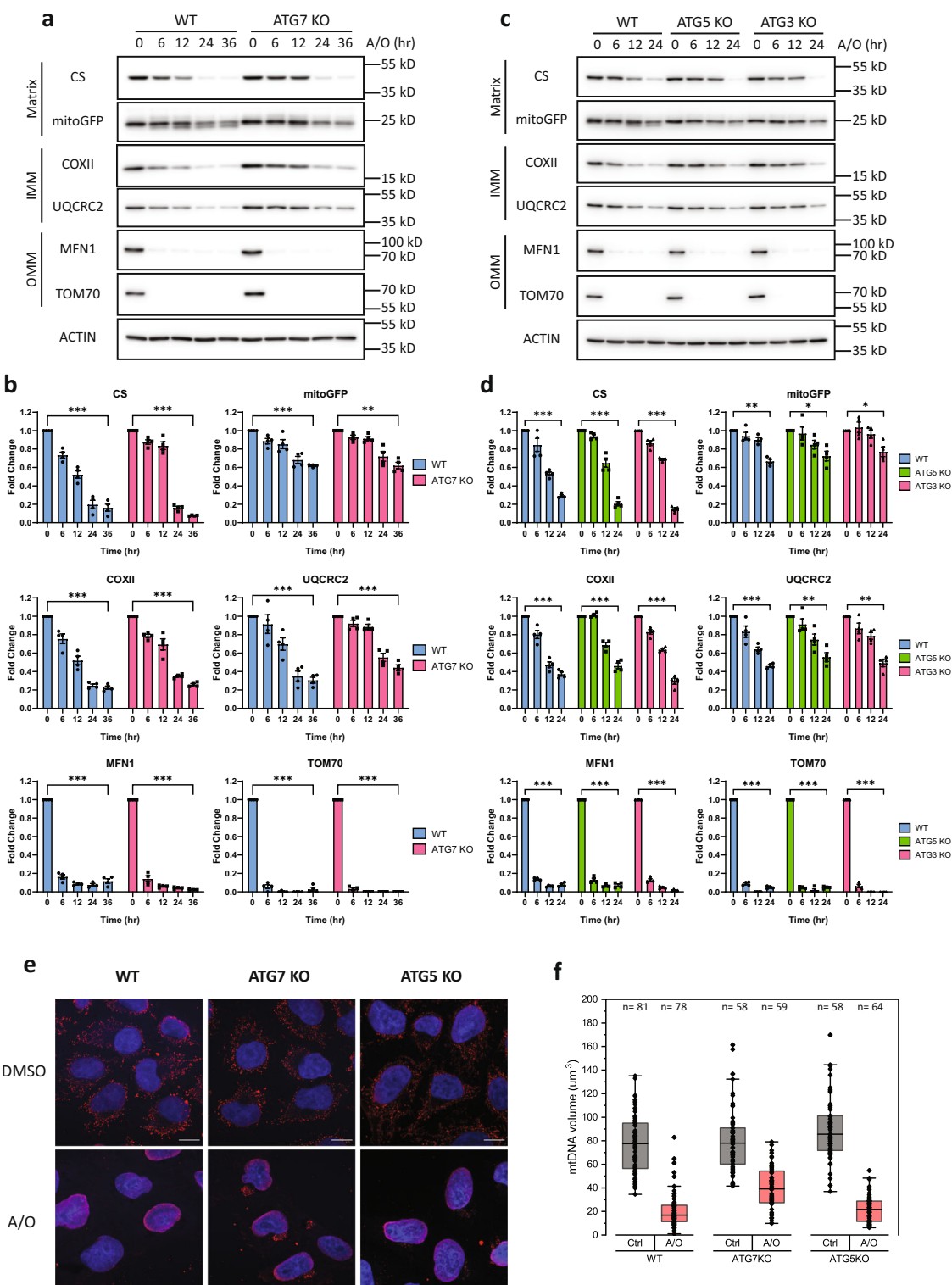

**Fig. 1 The mATG8-conjugation system is not required for PINK1–Parkin-mediated mitochondria clearance. a** WT or ATG7-KO HeLa stably expressing mCherry-Parkin were treated with antimycin A and oligomycin (A/O, 1 μM each) for the indicated duration. Mitochondria content was assessed by immunoblotting of proteins from different mitochondria compartments. **b** Quantification of mitochondrial protein changes from (**a**). Mean of $n = 4$ independent replicates ± SEM is shown. **c** WT, ATG5-KO, or ATG3-KO HeLa stably expressing mCherry-Parkin were treated with A/O for the indicated duration. **d** Quantification of mitochondrial protein changes from (**c**). Mean of $n = 4$ independent replicates ± SEM is shown. **e** Representative immunofluorescence images of WT, ATG7-KO and ATG5-KO cells stably expressing YFP-Parkin treated with A/O for 24 h and stained for mtDNA (red) and DAPI (blue). Scale bar = 10 μm. **f** Quantification of mtDNA volume from (**e**). Box plots indicate median (middle line), 25th, 75th percentile (box) and outlier limits (whiskers) with individual measurement data points overlaid. $P$ values in (**b**, **d**) were calculated by two-way ANOVA followed by Dunnett's multiple comparisons test against the 0 h timepoint. $^*P < 0.05$, $^{**}P < 0.01$, $^{***}P < 0.001$, and n.s. denotes not significant. See source data for exact $P$ values.

to GFP cleavage from GFP-LC3 during autophagy[55]) upon A/O treatment confirms that mitochondria were sent for lysosomal degradation as expected (Fig. 1a). It is known that OMM proteins such as MFN1 and TOM70 are rapidly degraded by the Ub-proteasome system independent of the autophagosome–lysosome pathway[56,57]. Consistently, we found similar patterns of OMM protein degradation in the WT and ATG7-KO cells (Fig. 1a, b). Unexpectedly, the degradation of IMM and matrix proteins was also observed in ATG7-KO cells (Fig. 1a, b), suggesting that mitochondria clearance during PINK1–Parkin mitophagy is not entirely dependent on the mATG8-conjugation system, which is responsible for mATG8 lipidation. To confirm this finding, we next generated ATG5 and ATG3 knockout cells, and as expected, these deletions effectively prevented the lipidation of the mATG8 orthologs LC3A, LC3B, GABARAP, and GABARAPL1 (Supplementary Fig. 1b). In line with our findings in ATG7-KO cells, similar patterns of mitochondria clearance were also observed in ATG5-KO or ATG3-KO cells upon mitophagy induction (Fig. 1c, d). Additionally, we performed immunostaining of mtDNA as an additional readout for mitochondria content. In support of our results, a significant reduction of mtDNA staining was observed in ATG7-KO and ATG5-KO cells upon A/O treatment, similar to that in WT cells (Fig. 1e, f).

**Clearance of damaged mitochondria requires key ATG components upstream of the mATG8-conjugation system**. We next sought to delineate the requirement of specific components upstream of the mATG8-conjugation system in mitochondria clearance. PINK1 initiates mitophagy by establishing the PINK1–Parkin feedforward loop on damaged mitochondria[5]. We therefore knocked out PINK1, which prevented the phosphorylation of Ub at Ser65 (a well-established PINK1-dependent phosphorylation site) upon A/O treatment (Fig. 2a and Supplementary Fig. 2a). As expected, mitochondria clearance was almost completely blocked in PINK1-KO cells treated with A/O (Fig. 2a and Supplementary Fig. 2b). On the other hand, TBK1 was not absolutely required for mitochondria clearance as the knockout of TBK1 in cells failed to prevent the degradation of various mitochondrial proteins (Fig. 2b and Supplementary Fig. 2c), which is consistent with previous reports[14,15]. It has been reported that ATG9A and the ULK1–FIP200 complex are recruited by OPTN and NPD52, respectively, to promote nascent autophagic membrane formation around damaged mitochondria[15,16]. We therefore investigated if ATG9A and the ULK1–FIP200 complex were required for mitochondria clearance. The degradation of mitochondrial markers was significantly inhibited in ATG9A-KO cells as compared to WT or ATG7-KO cells (Fig. 2c and Supplementary Fig. 2d). Additionally, the knockout of FIP200 in cells significantly impaired the degradation of mitochondrial proteins (Fig. 2d and Supplementary Fig. 2e). Consistently, inhibition of ULK1/2 by a chemical inhibitor SBI-0206965 also effectively prevented the degradation of IMM and matrix proteins (Fig. 2e). Together, our data suggest that key ATG components involved in autophagosome biogenesis upstream of the mATG8-conjugation system are still required for PINK1–Parkin-mediated mitochondria clearance.

Given that the lysosomal degradation of mitochondria represents the terminal step of mitophagy, we also assessed if mitochondria clearance is dependent on lysosomal function. Inhibition of lysosomal activity with lysosome inhibitors bafilomycin A1 (BafA1) or chloroquine (CQ) resulted in the partial impairment of IMM and matrix protein degradation induced by A/O, while the degradation of OMM proteins were unaffected (Fig. 2f and Supplementary Fig. 2f, g). Consistent with previous reports[56–58], OMM proteins were degraded in a proteasome-dependent manner as the addition of proteasome

inhibitors MG132 or epoxomicin completely blocked the degradation of OMM proteins (Fig. 2f and Supplementary Fig. 2h). A partial blockage of IMM protein degradation upon proteasomal inhibition was also observed (Fig. 2f and Supplementary Fig. 2h), which was in-line with previous studies showing the requirement of the proteasome in mitophagy through the proteasomal degradation of mitochondria–ER tethers[56,58,59].

Since normal autophagic activity has been demonstrated in ATG5- or ATG7-independent autophagy[36,37], we next asked if the mitochondria clearance we observed still relied on the autophagosome–lysosome pathway. Interestingly, BafA1 treatment did not block the degradation of IMM and matrix proteins in ATG7-KO cells (Fig. 2f and Supplementary Fig. 2f), demonstrating that ATG7-independent mitochondria clearance did not occur via lysosomal degradation. On the other hand, proteasome inhibition blocked the degradation of proteins from all compartments (Fig. 2f). We further confirmed this finding in ATG5-KO and ATG3-KO cells and obtained consistent results (Supplementary Fig. 2i). To further evaluate if mitochondria are sent to the lysosomal compartments for degradation in the absence of ATG7, we performed immunostaining of HSP60 and LAMP2 to label mitochondria and lysosomes, respectively. Indeed, the formation of LAMP2 rings engulfing HSP60 puncta were significantly reduced in ATG7-KO cells when compared to WT cells (Fig. 2g, h), thus suggesting that ATG7-independent mitochondria clearance does not occur via lysosomal degradation.

**Damaged mitochondria are secreted via a pathway distinct from small EVs**. Mitochondria have been shown to be extruded from multiple cell types during different stress conditions[45–47]. We therefore considered the possibility that mATG8 conjugation system-independent mitochondria clearance occurs via the unconventional secretion of damaged mitochondria. To assess this, we first performed SILAC-based quantitative proteomics comparing the extracellular vesicle (EV) fraction isolated from A/O treated WT and ATG7-KO cells (Fig. 3a). Overall, 1086 proteins were identified to have SILAC ratios in both forward (WT heavy:KO light) and reverse (WT light: KO heavy) labelling experiments (Fig. 3b and Supplementary Data 1). Further analysis revealed that 76 proteins had more than 1.5-fold enrichment when comparing ATG7-KO to WT in both forward and reverse labelling experiments (Fig. 3c and Supplementary Data 1). Gene ontology (GO) cellular compartment analysis of these 76 proteins revealed a statistically significant enrichment from the mitochondrial and endolysosomal compartments (Fig. 3b, d), suggesting that ATG7-KO cells secrete more mitochondrial and lysosomal contents than WT cells under mitochondria damaging conditions.

To further validate this finding, we next probed for mitochondrial markers from the EV fraction isolated from the conditioned media of WT or ATG7-KO cells. When compared to treated WT cells, the EV fraction from treated ATG7-KO cells were found to have higher levels of Ser65 p-Ub (a marker of damaged mitochondria), IMM and matrix proteins upon mitochondria damage (Fig. 4a), suggesting increased secretion of mitochondria in ATG7-KO cells. An increase in mitochondrial markers in the EV fraction were also observed from A/O-treated ATG5-KO or ATG3-KO cells (Supplementary Fig. 3a), further demonstrating increased secretion of mitochondria when the mATG8-conjugation system is defective. The changes in mitochondria secretion were not attributed to non-specific leakage from cells, as a LDH release assay revealed no differences between WT, ATG7-KO, ATG5-KO, or ATG3-KO cells (Supplementary Fig. 3b). TEM analysis of the extracellular region of

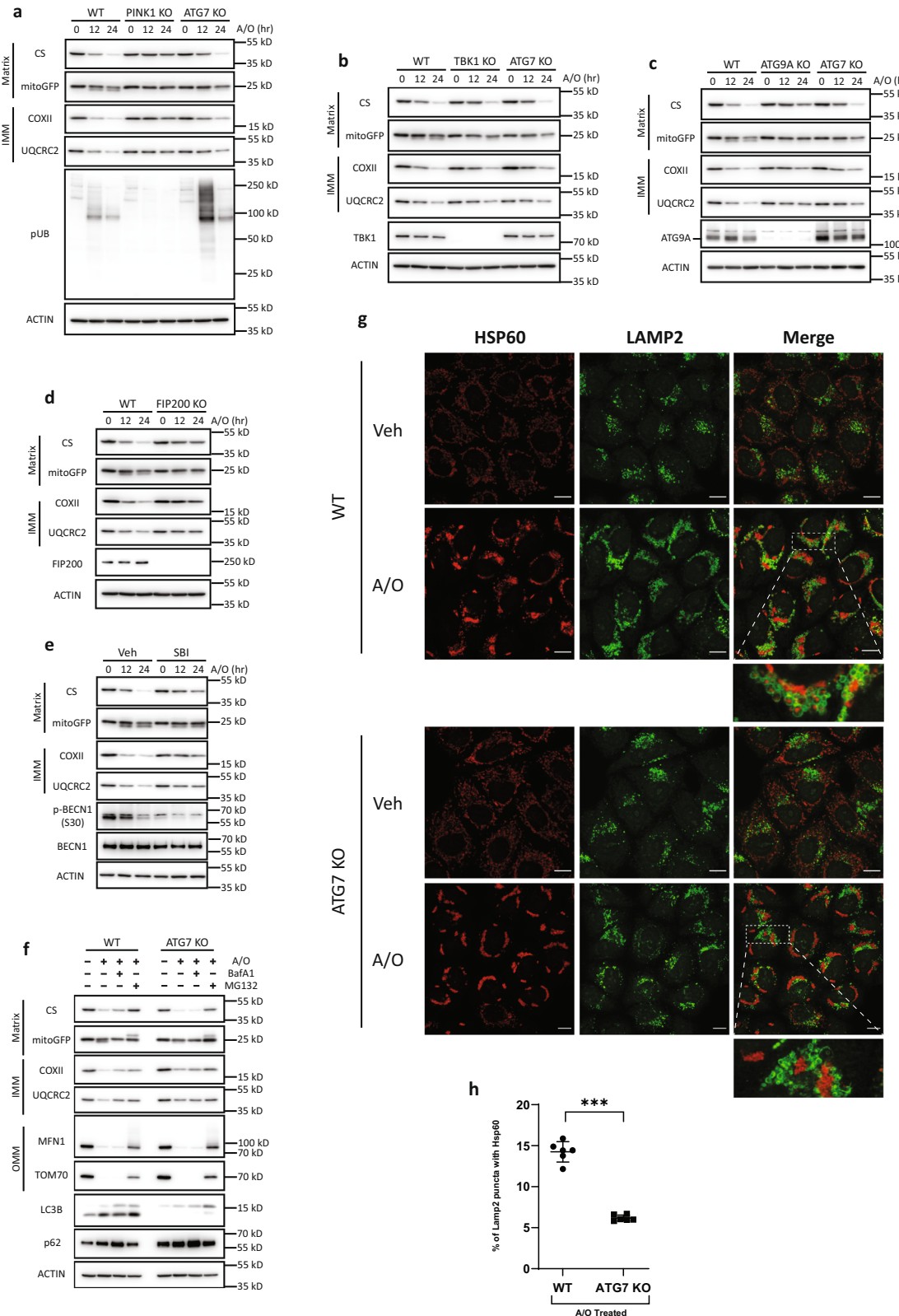

treated ATG7-KO also revealed single-membrane vesicular structures that resembled mitochondria with ruptured outer membranes (Supplementary Fig. 3c), although we were unable to identify the nature of these vesicles.

Moreover, we confirmed these findings using an in vivo exhaustive exercise (EE) model to acutely stress mitochondria and activate the PINK1–Parkin pathway in vivo, as previously described[60]. To assess the extracellular release of mitochondria, WT or muscle-specific ATG7-KO mice (ATG7-mKO) were subjected to EE followed by isolation of EVs from serum. In response to EE, ATG7-mKO mice displayed a significantly higher level of mitochondrial markers from the serum EV fraction, while no appreciable increase was observed in WT mice (Fig. 4b). Overall, defects in the mATG8-conjugation system results in the

**Fig. 2 Clearance of damaged mitochondria requires key ATG components upstream of the mATG8-conjugation system. a** WT, ATG7-KO, and PINK1-KO, **b** TBK1-KO, **c** ATG9A-KO, **d** FIP200-KO HeLa cells stably expressing mCherry-Parkin were treated with A/O for the indicated duration. Mitochondria content was assessed by immunoblotting of proteins from different mitochondria compartments. **e** WT HeLa stably expressing mCherry-Parkin was treated with A/O, with or without SBI- 0206965 (25 μM). **f** WT and ATG7-KO cells stably expressing mCherry-Parkin were treated with A/O for 24 h, with or without Bafilomycin A1 (BafA1, 200 nM) or MG132 (10 μM) to inhibit lysosomal or proteasome degradation, respectively. **g** Representative immunofluorescence images of WT and ATG7-KO cells treated with A/O for 6 h and stained for HSP60 (red) and LAMP2 (green). Scale bar = 10 μm. **h** Quantification of A/O-treated groups in **g** showing percentage of LAMP2 puncta colocalised with HSP60. Mean of $n = 6$ fields of view ± SEM is shown. $P$ values were calculated by two-tailed Student's $T$ test. *$P < 0.05$, **$P < 0.01$, ***$P < 0.001$, and ns denotes not significant. See source data for exact $P$ values.

secretion of mitochondria in response to mitochondria damage/stress both in vitro and in vivo.

We next attempted to elucidate the route by which mitochondria are secreted. Since the small EV markers CD63 and CD9[61,62] were enriched in the EV fraction from ATG7-KO cells treated with A/O (Fig. 4a), we asked if damaged mitochondria were secreted via small EVs. Nanoparticle tracking analysis revealed that A/O-treated ATG7-KO or ATG5-KO cells produced more small EVs that were ~50–150 nm in size (Fig. 4c and Supplementary Fig. 3d). We next performed nPLEX analysis, which has been previously developed to immunocapture CD63+ small EVs and probe their molecular cargos[63,64]. The assay revealed a significant increase in signals of CD63 and the small EV cargo HSP70 from the EV fraction of treated ATG5-KO cells (Supplementary Fig. 3e). However, signals of Parkin, fumarase and Ser65 p-Ub were not detected in CD63+ small EVs, suggesting that damaged mitochondria were unlikely to be enveloped by CD63+ small EVs (Supplementary Fig. 3e). To confirm this notion, we performed size-exclusion filtration of the conditioned media from treated ATG7-KO cells before the isolation of EVs. Filtration (0.22 μm cut-off) of conditioned media would deplete larger particles (e.g. fragmented mitochondria) but not small EVs such as exosomes and small ectosomes, which are smaller than the filter cutoff. As shown in Fig. 4d, filtration depleted the EV fraction of mitochondrial markers, but not CD9, again suggesting that the extracellular mitochondria are a distinct population from small EVs. Additionally, proteinase K (PK) protection assay of the EV fraction also revealed that unlike the small EV cargo HSP70, mitochondria contents were not protected from PK-mediated degradation, further suggesting that the secreted mitochondria were not present within small EVs (Fig. 4e). To further confirm this, we separated the EV fraction isolated from A/O-treated ATG7-KO cells by a iodixanol-based density gradient. Mitochondrial markers ACO2 and COXII were not found in the same fractions as the small EV marker CD9, demonstrating that secreted mitochondria and small EVs were distinct populations (Fig. 4f).

**Mitochondria secretion occurs via secretory autophagy.** Several reports have shown that mATG8s and its conjugation system are not required for autophagosome formation[33–35]. We therefore tested the possibility that mitochondria are secreted via secretory autophagy, a pathway which utilizes the autophagosome as the vesicular carrier for unconventional secretory cargos[65]. First, we determined if complete mitophagosomes can still form in cells with a defective mATG8-conjugation system. To do this, NDP52, a well-defined mitophagy cargo receptor, was used to determine the levels of mitophagosome-protected cargos, while ATG9A served as a control that is not incorporated into autophagosomes. As shown in Fig. 5a, NDP52 was protected from proteinase K-mediated degradation in homogenates from both WT and ATG7-KO cells treated with A/O and BafA1, while ATG9A was readily degraded by proteinase K. Notably, A/O treatment markedly enhanced the level of NDP52 resistant to proteinase K

in both WT and ATG7-KO cells, although to a lesser degree in the ATG7-KO cells (Fig. 5a, b), which is in-line with an earlier report in mATG8 hexa KO cells[33]. Consistently, addition of the class III PI3K inhibitor SAR405 to prevent the formation of autophagosome membranes abolished the protection of NDP52 against PK degradation, suggesting that mitochondria are indeed sequestered by intact autophagosomes in both WT and ATG7-KO cells (Fig. 5a, b). This finding was also corroborated by TEM analysis showing the presence of mitophagosomes in both A/O-treated WT and ATG7-KO cells (Fig. 5c).

Since secretory autophagy uses the autophagosome as the cargo carrier, we next asked if inhibiting autophagosome formation prevents the secretion of mitochondria. We first deleted ATG14 or FIP200 in ATG7-KO cells and confirmed the lack of nascent autophagosome membranes by immunostaining of the phagophore marker WIPI2. WIPI2 puncta that colocalized with mitochondria were observed in A/O-treated ATG7-KO cells, but were greatly reduced in ATG14/ATG7 or FIP200/ATG7 double knockout cells (Supplementary Fig. 4a, b). Concomitantly, deletion of ATG14 or FIP200 in ATG7-KO cells abolished the presence of mitochondrial markers and Ser65 p-UB, but not CD9, in the EV fraction upon mitochondria damage (Fig. 5d, e). Knocking out ATG9A, which has been shown to mediate autophagosome formation, in ATG7-KO cells also blocked mitochondria secretion, in support of our notion that mitochondria are extruded via secretory autophagy (Fig. 5f). Consistently, deletion of the SNARE protein SNAP23, which has been previously shown to mediate autophagosome-plasma membrane fusion during secretory autophagy[51], also reduced mitochondria secretion in ATG7-KO cells (Fig. 5g).

Given that the fate of autophagosomes bifurcates into fusion with lysosomes in degradative autophagy or with the plasma membrane in secretory autophagy, we next examined if the inhibition of autophagosome–lysosome fusion would promote the autophagic secretion of mitochondria. The small GTPase RAB7A has been shown to mediate the fusion of lysosomes with autophagosomes or endosomes[66]. Deletion of RAB7A markedly increased the secretion of mitochondria markers under basal conditions, while the secretion of mitochondria upon mitochondria damage was not affected (Fig. 5h). Interestingly, the EV marker CD9 was significantly higher under both basal and A/O-treated conditions (Fig. 5h). Taken together, we present evidence suggesting that damaged mitochondria can be extruded from cells via a secretory autophagy pathway, a process independent of ATG7 that mediates mATG8 lipidation, which we term as Autophagic Secretion of Mitochondria (ASM).

**Suppression of ASM impairs mitochondria clearance.** Since ASM could serve as an alternate mitochondria clearance pathway to remove dysfunctional mitochondria, we next tested if the inhibition of ASM perturbs mitochondria clearance. When compared to ATG7-KO cells, the degradation of IMM and matrix proteins were significantly inhibited in ATG14/ATG7-DKO or FIP200/ATG7-DKO cells upon A/O treatment (Fig. 6a, b and

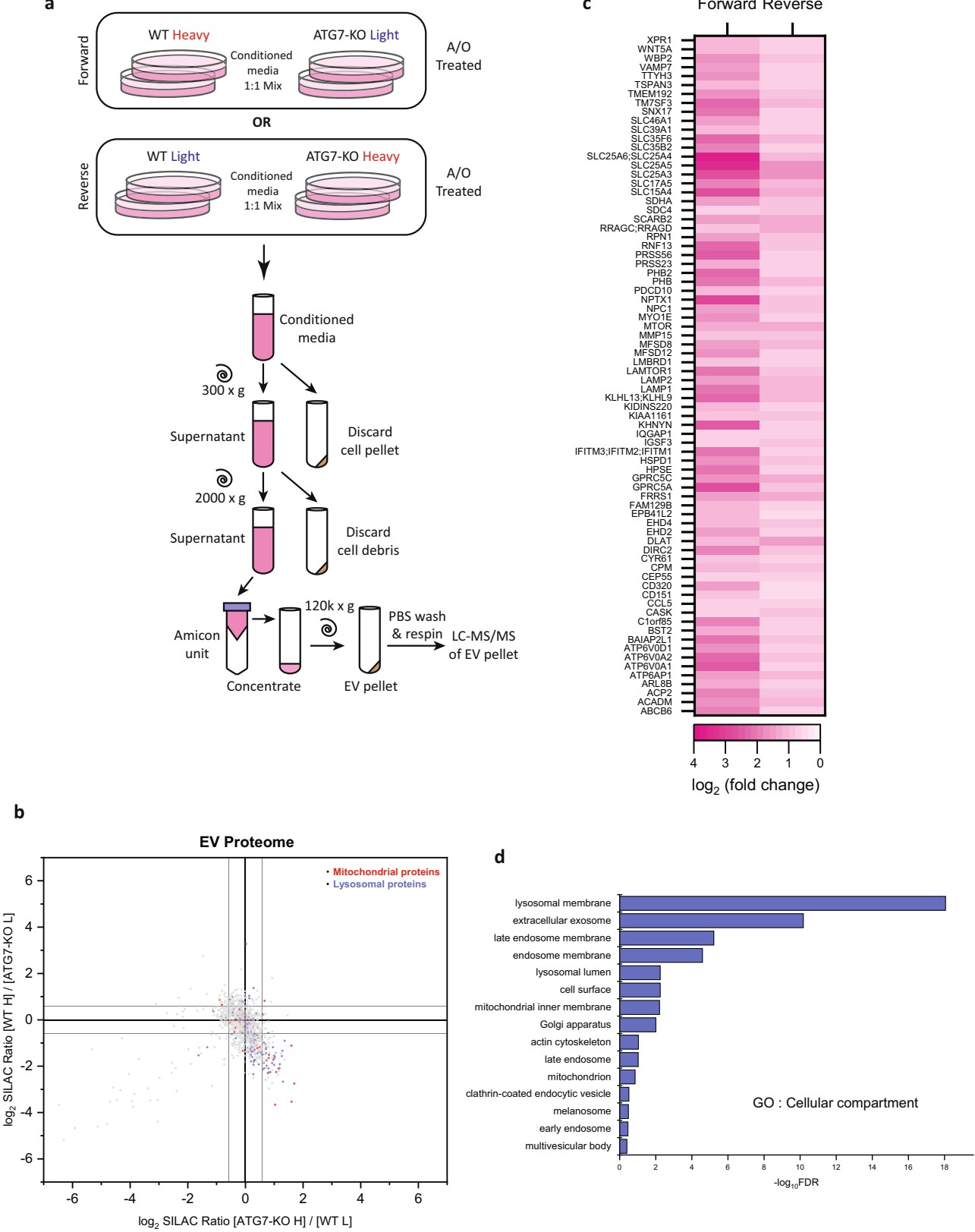

**Fig. 3 Global proteomics profiling of extracellular vesicles. a** Schematic diagram of SILAC labelling and extracellular vesicles (EV) isolation. **b** log$_2$[H/L] dotplot of EV proteins that had SILAC ratios in forward (WT heavy; ATG7-KO light) and reverse (ATG7-KO heavy; WT light) labelling experiments. Grey lines denote a 1.5× fold enrichment cutoff. **c** Heatmap of proteins identified to have a >1.5 fold enrichment when comparing ATG7-KO to WT EVs in both forward and reverse labelling experiments. **d** Gene ontology analysis (cellular compartment) of proteins that were enriched >1.5 fold in EVs of ATG7-KO vs. WT, plotted according to −log$_{10}$FDR.

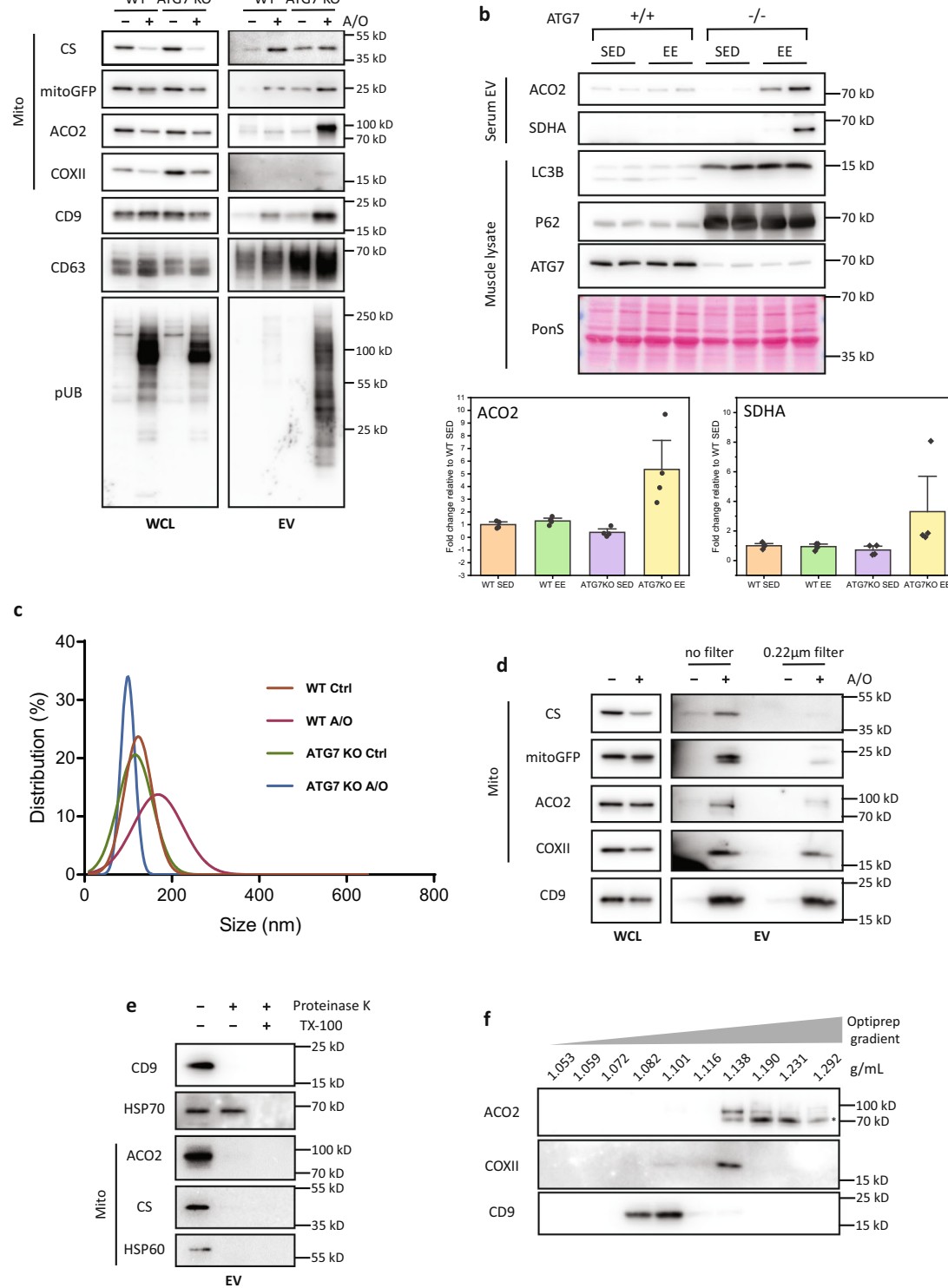

**Fig. 4 Damaged mitochondria are secreted via a pathway distinct from small EVs. a** Conditioned media from WT and ATG7-KO HeLa stably expressing mCherry-Parkin treated with A/O for 24 h were collected. Extracellular vesicles (EVs) were isolated by differential ultracentrifugation and immunoblotted for mitochondria and small EV markers. **b** Control or muscle specific ATG7-KO (*Atg7f/f;Ckmm-cre*) mice were subjected to 3 consecutive days of exhaustive exercise. Serum EVs or TA muscle tissue were then harvested. Representative immunoblots and quantification of the mitochondria markers (ACO2 and SDHA) from serum EVs are shown. Mean of $n = 4$ mice per group ±SEM. **c** Nanoparticle tracking analysis of EVs from WT and ATG7-KO cells untreated or treated with A/O for 24 h. **d** Conditioned media from 24 h A/O treated ATG7-KO HeLa stably expressing mCherry-Parkin were filtered (0.22 μm cutoff) before isolation of EVs by differential ultracentrifugation and immunoblotting. **e** Proteinase K protection assay of EVs isolated by differential ultracentrifugation from 24 h A/O treated ATG7-KO HeLa stably expressing mCherry-Parkin. **f** EVs from 24 h A/O treated ATG7-KO cells were separated by a 5–40% bottom-up iodixanol density flotation gradient and subjected to immunoblotting. *Non-specific bands.

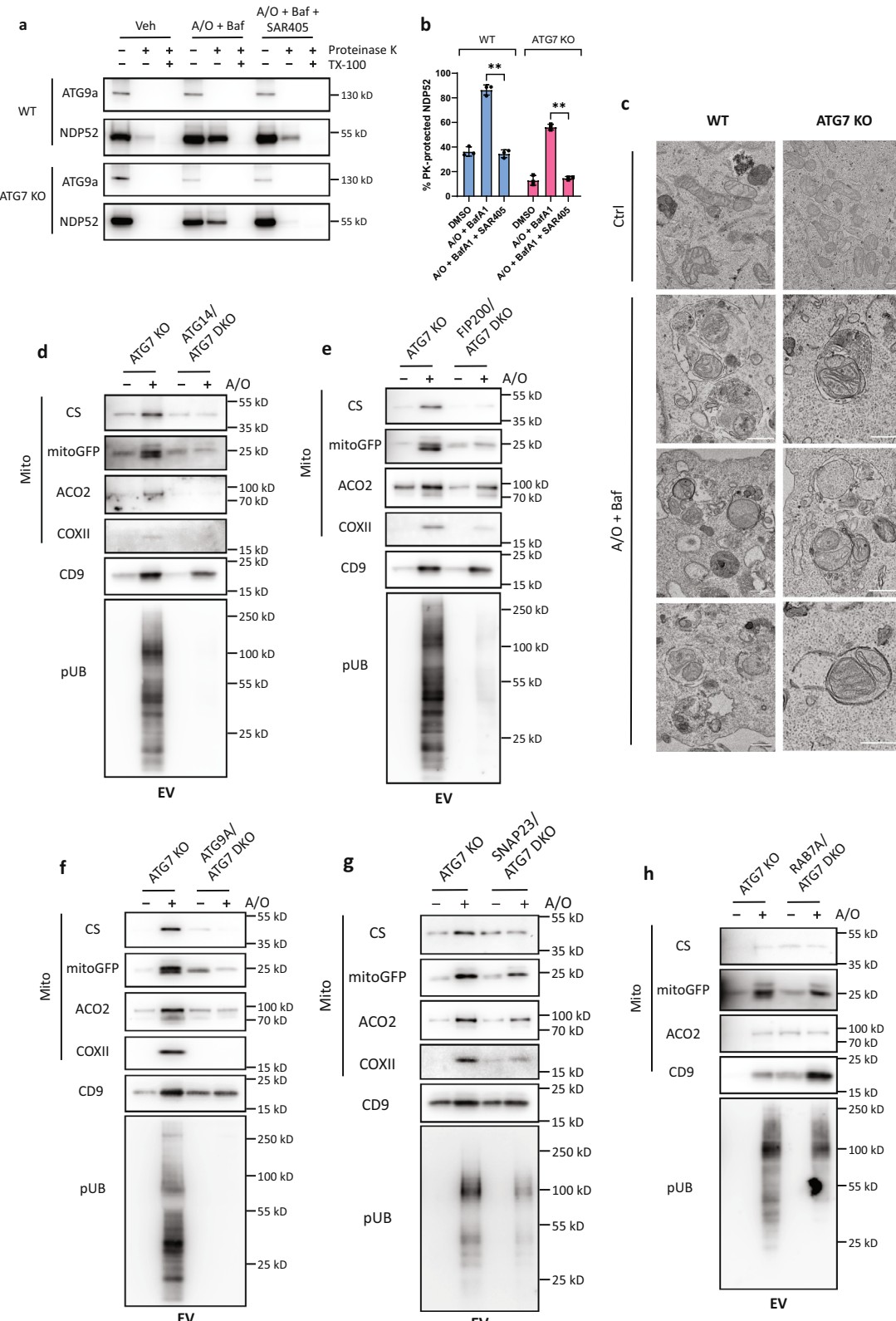

Supplementary Fig. 5a, b). Consistently, inhibition of ULK1 with SBI-0206965 prevented mitochondria clearance in ATG7-KO cells treated with A/O (Fig. 6c). Additionally, the degradation of mitochondrial markers upon A/O treatment was significantly inhibited when ATG9A was deleted in ATG7-KO cells (Fig. 6d and Supplementary Fig. 5c). Conversely, the deletion of RAB7A did not affect mitochondria degradation in ATG7-KO cells

(Fig. 6e and Supplementary Fig. 5d). Together, these results suggest that ASM exists as a MQC pathway to clear dysfunctional mitochondria when the mATG8-conjugation system is defective.

**The innate immune cGAS–STING pathway is potently activated by ASM.** Mitochondrial DNA (mtDNA) is a potent

**Fig. 5 ATG7-independent mitochondria secretion occurs via secretory autophagy. a** Proteinase K protection assay of homogenates from WT or ATG7-KO cells stably expressing mCherry-Parkin treated with A/O and BafA1 (200 nM) for 12 h in the presence or absence of SAR405 (5 μM). **b** Quantification of the percentage of NDP52 protected from proteinase K digestion for each treatment condition. Mean of $n = 3$ independent replicates ±SEM is shown. $P$ values were calculated by two-way ANOVA followed by Dunnett's multiple comparisons test against the A/O + BafA1 group. $*P < 0.05$, $**P < 0.01$, $***P < 0.001$, and ns denotes not significant. See source data for exact $P$ values. **c** Representative TEM images of mature autophagosomes containing mitochondria from WT or ATG7-KO cells stably expressing mCherry-Parkin treated with A/O + BafA1 for 16 h. Scale bar represents 500 nm. **d** ATG7-KO and ATG14/ATG7 DKO; **e** ATG7-KO and FIP200/ATG7 DKO; **f** ATG7-KO and ATG9A/ATG7 DKO; **g** ATG7-KO and SNAP23/ATG7 DKO; **h** or ATG7-KO and RAB7A/ATG7 DKO HeLa stably-expressing mCherry-Parkin were treated with A/O for 24 h. Extracellular vesicles (EVs) were isolated by differential ultracentrifugation and immunoblotted for the indicated proteins.

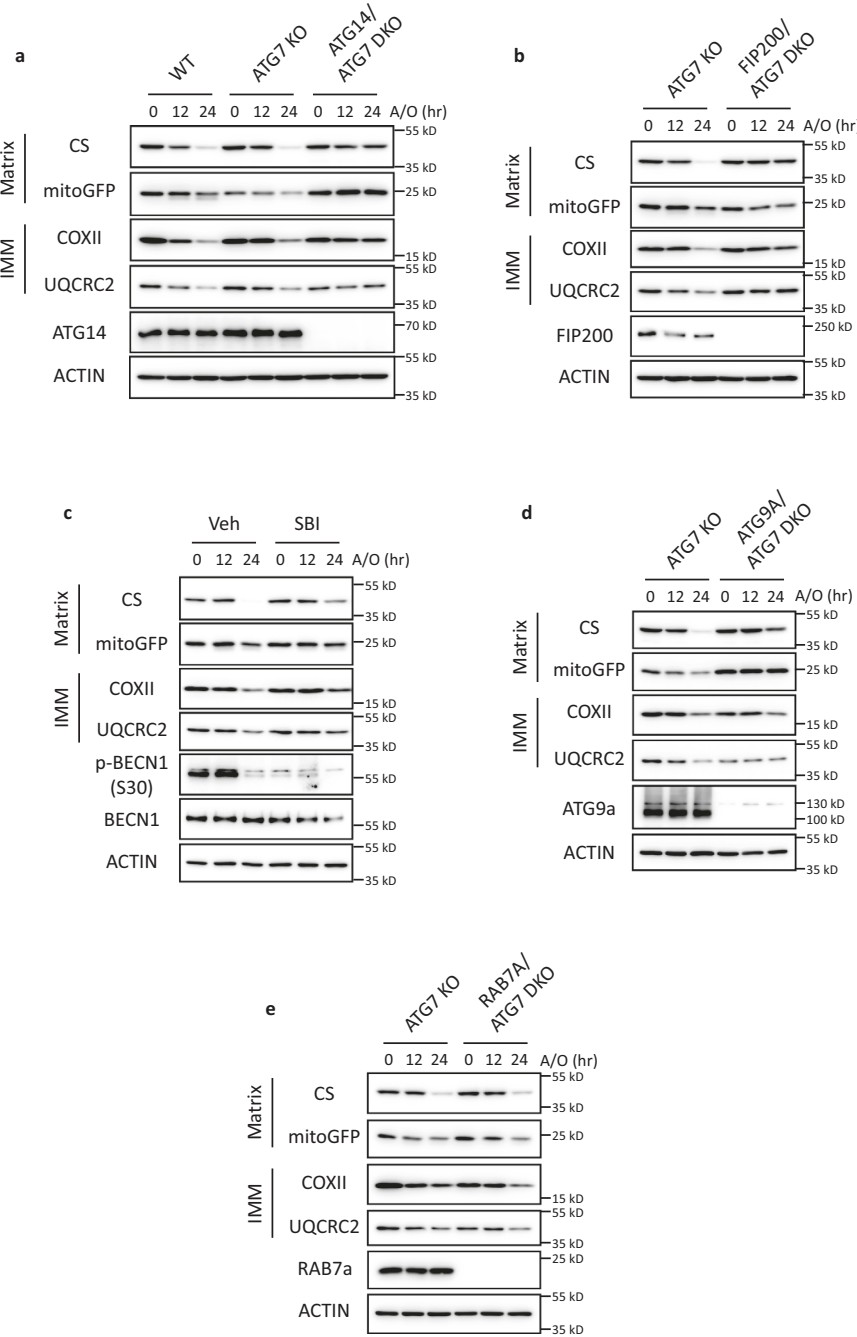

**Fig. 6 ATG7-independent mitochondria clearance is impaired when autophagic secretion of mitochondria is inhibited. a** WT, ATG7-KO, and ATG14/ATG7 DKO HeLa stably expressing mCherry-Parkin were treated with A/O for the indicated duration and immunoblotted for proteins from different mitochondria compartments. **b** ATG7-KO and FIP200/ATG7 DKO cells were treated with A/O for the indicated duration. **c** ATG7-KO cells stably expressing mCherry-Parkin was treated with A/O, with or without SBI- 0206965 (25 μM). **d** ATG7-KO and ATG9A/ ATG7 were treated with A/O for the indicated duration. **e** ATG7-KO and RAB7A/ATG7 DKO cells were treated with A/O for the indicated duration.

damage-associated molecular pattern (DAMP) that has been reported to activate the cGAS–STING innate immune pathway[67–69]. mtDNA has also been linked to immune responses and the pathology of neurodegenerative disorders when the mitophagy machinery is defective[60]. To address whether secreted mitochondria are capable of activating the innate immune response, HeLa cells were incubated with EVs isolated from A/O-treated WT or ATG7-KO cells. The addition of EVs from ATG7-KO cells resulted in higher levels of cGAS–STING activation as measured by significantly higher levels of p-TBK1 (S172), p-STING (S366), and p-IFR3 (S386), in comparison to the addition of EVs from WT cells (Fig. 7a, b), consistent with increased ASM when the mATG8-conjugation system is defective. Concomitantly, the expression of STING-dependent proinflammatory genes (*IL6*, *IL8*, *IFNB1*, *CXCL10*, *IFIT1*) was significantly elevated in HeLa cells treated with EVs derived from ATG7-KO cells as compared to those derived from WT cells (Fig. 7c). The secretion of IL6 into the culture media was also significantly higher when recipient cells were incubated with EVs derived from ATG7-KO than those from WT cells (Fig. 7d). Moreover, we confirmed that the extracellular IL6 was indeed secreted from recipient cells and not derived from the EVs per se (Fig. 7h, see no cell control group). Lastly, we assessed the effect of a dampened ASM response on cGAS–STING activation. We found that the expression of STING-dependent genes and IL6 secretion from recipient cells were markedly lower when incubated with EVs isolated from ATG14/ATG7-DKO cells as compared to ATG7-KO cells (Figs. 7e, f), in line with our expectations of significantly diminished cGAS–STING activation when ASM is inhibited. Consistently, EVs from PINK1/ATG7-DKO cells had no effect on the expression STING-dependent genes or IL6 secretion from recipient cells (Fig. 7g, h).

## Discussion

Here, we present a MQC pathway that relies on secretory autophagy, a process we termed as ASM (Fig. 8). Firstly, we found that mitochondria could be cleared from cells in the absence of the mATG8-conjugation machinery during PINK1–Parkin mitophagy, but was still dependent on upstream ATG proteins involved in autophagosome biogenesis. Secondly, mitochondria clearance in the absence of the mATG8-conjugation system was found to be attributed to its extracellular release via secretory autophagy. Lastly, ASM was found to elicit a pro-inflammatory response in recipient cells via activation of the cGAS–STING pathway.

In search of uncharacterized regulatory mechanisms in control of mitophagy, we initially made the unexpected observation that mitochondria could be cleared independent of the mATG8-conjugation system essential for mATG8 lipidation during PINK1–Parkin-dependent mitophagy (Fig. 1). Our findings are similar with previous studies in which ATG5 or ATG7 was found to be dispensable in the clearance of mitochondria during erythropoiesis, iPSC reprogramming or energy stress[36,38–40]. Notably, these models of mitochondria clearance were independent of PINK1 and Parkin, suggesting that mitochondria may undergo mATG8 lipidation-independent autophagic secretion regardless of the mitophagy pathway activated.

mATG8s have been shown to be required for the clearance of mitochondria during PINK1–Parkin mitophagy by mediating late-stage autophagic events such as autophagosome–lysosome fusion[33,70]. In contrast, our results revealed that mitochondria clearance could occur without the mATG8-conjugation system components such as ATG7, ATG5, or ATG3, hinting towards a difference in function of lipidated and unlipidated-mATG8s. Indeed, it has been reported that autophagosome–lysosome

fusion during autophagy was found to be normal in ATG3 KO MEFs, although other defects including impaired autophagosome closure and inner autolysosomal membrane degradation were observed[34]. Non-canonical recruitment of lipidated LC3 to single-membrane organelles such as lysosomes, phagosomes, or endosomes have also been reported[71–74], further highlighting the complexity in mATG8s function.

We unexpectedly found that mATG8 lipidation-independent mitochondria clearance did not rely on lysosomal degradation (Fig. 2f), which prompted us to search for other modes of mitochondria clearance. In this study, we showed that mitochondria were shed via a pathway distinct from small EVs when mATG8 lipidation is deficient (Fig. 4). This finding was surprising since studies have reported the presence of mitochondrial content in different types of small EVs released from multiple cell types[48,75,76]. Instead, we found that mitochondria were extruded via secretory autophagy, a process that relies on the PINK1–Parkin model of mitophagosome formation as the ablation of key autophagosome biogenesis ATGs effectively prevented the secretion of mitochondria. Consistently, previous studies have shown that mATG8s are not required for the formation of mitophagosomes, as autophagy adaptor proteins such as NDP52 and OPTN can directly recruit the autophagy machinery to nucleate nascent autophagic membranes on targeted mitochondria[15,16,33]. Our findings also tend to suggest that the extracellular mitochondria were not enveloped by a vesicle which would presumably be derived from the inner autophagosomal membrane. Interestingly, similar findings were also observed in previous secretory autophagy studies, as the soluble cargos of interest in those studies could be directly measured from the conditioned media[51,53,77]. Thus, how the secretory autophagosome fuses with the plasma membrane, or whether degradation of the inner autophagosome membrane occurs during secretory autophagy remain key questions to be answered in future studies.

Given that mATG8s mediate autophagosome–lysosome fusion, it is conceivable that the deficiency of mATG8 lipidation may redirect autophagosome trafficking towards the plasma membrane instead of lysosomes. Interestingly, previous studies have demonstrated the requirement of ATG5 (which is required for mATG8 lipidation) in secretory autophagy[51,53,54], which is contrary to our study. One possibility for this discrepancy may be due to the different requirements of mATG8s in the selective recruitment of specific cargos into autophagosomes for secretion. Another possibility is that secretory autophagy is regulated by a non-canonical function of the ATG12-ATG5 conjugate that is unrelated to mATG8 lipidation. Thus far, secretory autophagy has only been widely known to be responsible for the unconventional secretion of leaderless protein cargoes including IL-1β, ferretin, and HMGB1[50–53].

Although both mitophagy and ASM results in the eventual clearance of mitochondria, the pathophysiological implication of these two processes are vastly different. PINK1–Parkin mitophagy has been previously reported to limit the release of damage-associated molecular patterns (DAMPs) caused by mitochondria stress in mice, which restrains STING-mediated inflammation and dopaminergic neuron cell death[60]. Additionally, microglial mitophagy inhibits neuroinflammation in β-amyloid and tau models of Alzheimer's disease[78], demonstrating that defective mitophagy contributes to the etiology of neurodegenerative diseases. Conversely, our data revealed an activation of the cGAS–STING pathway and secretion of pro-inflammatory cytokines when mitochondria are cleared by ASM in the absence of ATG7 (Fig. 7), although it is remains to be confirmed if mtDNA was indeed responsible for this phenotype. Intriguingly, while multiple studies have demonstrated the ability of extracellular mitochondria or mtDNA-containing EVs to activate cGAS–STING signalling[79–81],

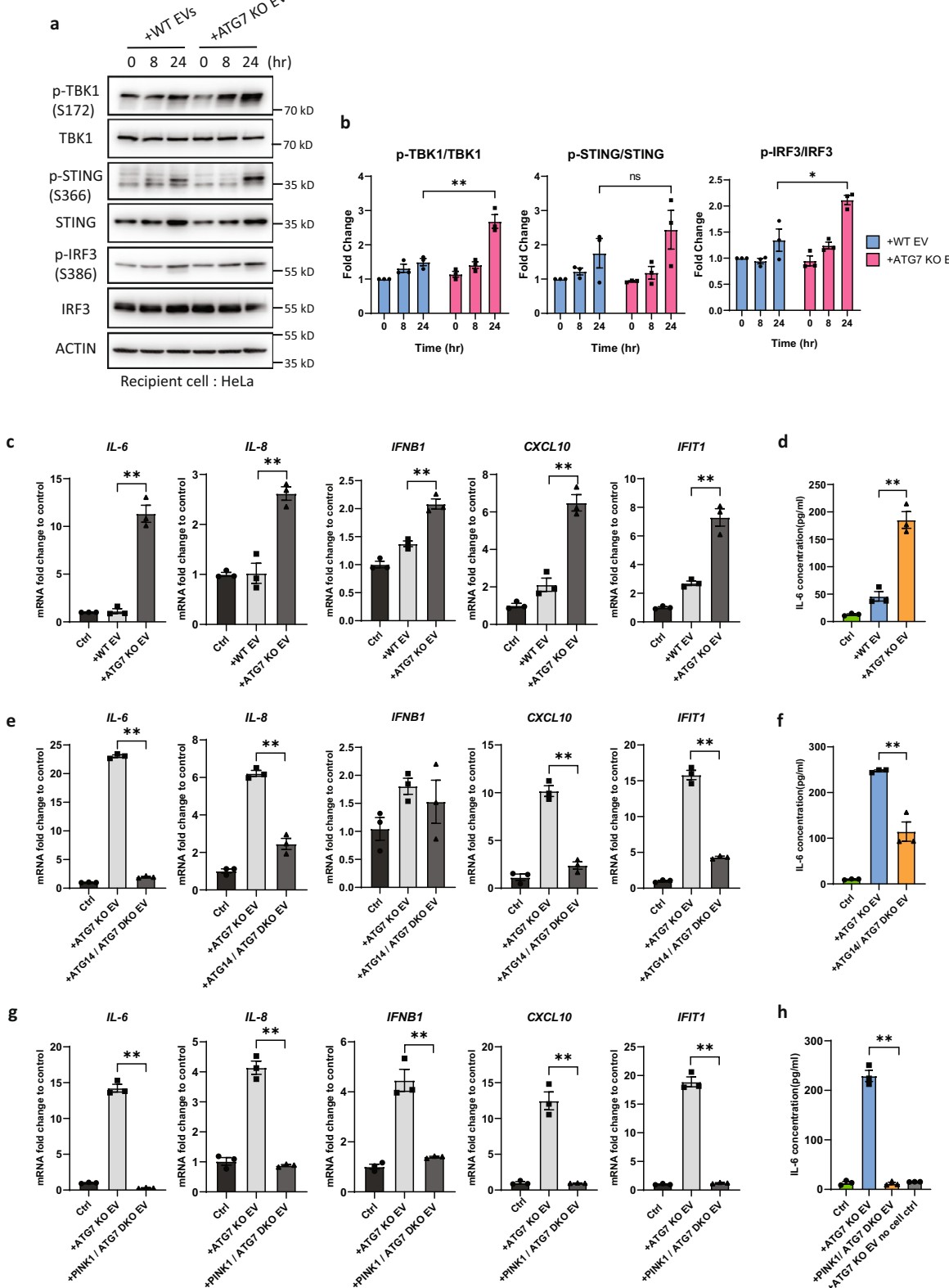

it remains unclear how mtDNA inside those EVs contacts cytosolic cGAS upon entry into its recipient cells. At present, it is known that neuronal inflammation and injury in neurodegenerative models involving neurotoxic proteins is attributed to the release of fragmented dysfunctional mitochondria into the neuronal milieu[49]. Similarly, impaired autophagosome–lysosome fusion and the accumulation of immature autophagic vacuoles are often

observed in these neurodegenerative diseases[82]. Thus, STING-mediated neuroinflammation in these diseases may be attributed to the diversion of dysfunctional mitochondria towards autophagic secretion instead of degradation. Additionally, mutations in ATG7 or ATG5 have been identified in neuronal pathologies such as Huntington's disease and Parkinson's disease[83–86], further supporting the notion that ASM upon defective mATG8 lipidation

**Fig. 7 cGAS-STING is potently activated by secreted mitochondria. a** Extracellular vesicles (EVs) isolated from A/O-treated WT or ATG7-KO HeLa stably expressing mCherry-Parkin were added to recipient HeLa cells for the indicated time. Activation of the cGAS–STING pathway was then assessed by immunoblotting. **b** Quantification of protein changes from **a**. Mean of $n = 3$ independent replicates ±SEM is shown. $P$ values were calculated by two-tailed Student's $T$ test. *$P < 0.05$, **$P < 0.01$, ***$P < 0.001$, and ns denotes not significant. See source data for exact $P$ values. **c, d** EVs isolated from A/O treated WT or ATG7-KO cells were added to recipient HeLa cells for 24 h. mRNA expression of STING-dependent inflammatory cytokines (**c**) or IL-6 secretion into culture media (**d**) was then assessed. **e, f** EVs isolated from A/O-treated ATG7-KO or ATG14/ATG7 DKO cells were added to recipient HeLa cells for 24 h. mRNA expression of STING-dependent inflammatory cytokines (**e**) or IL-6 secretion into culture media (**f**) was then assessed. **g, h** EVs isolated from A/O treated ATG7-KO or PINK1/ATG7 DKO cells were added to recipient HeLa cells for 24 h. mRNA expression of STING-dependent inflammatory cytokines (**g**) or IL-6 secretion into culture media (**h**) was then assessed. No cell control group in **h** denotes the addition of isolated ATG7-KO EVs into wells with no recipient HeLa cells. Mean of $n = 3$ independent replicates ±SEM is shown. $P$ values were calculated by two-tailed Student's $T$ test. *$P < 0.05$, **$P < 0.01$, ***$P < 0.001$, and ns denotes not significant. See source data for exact $P$ values.

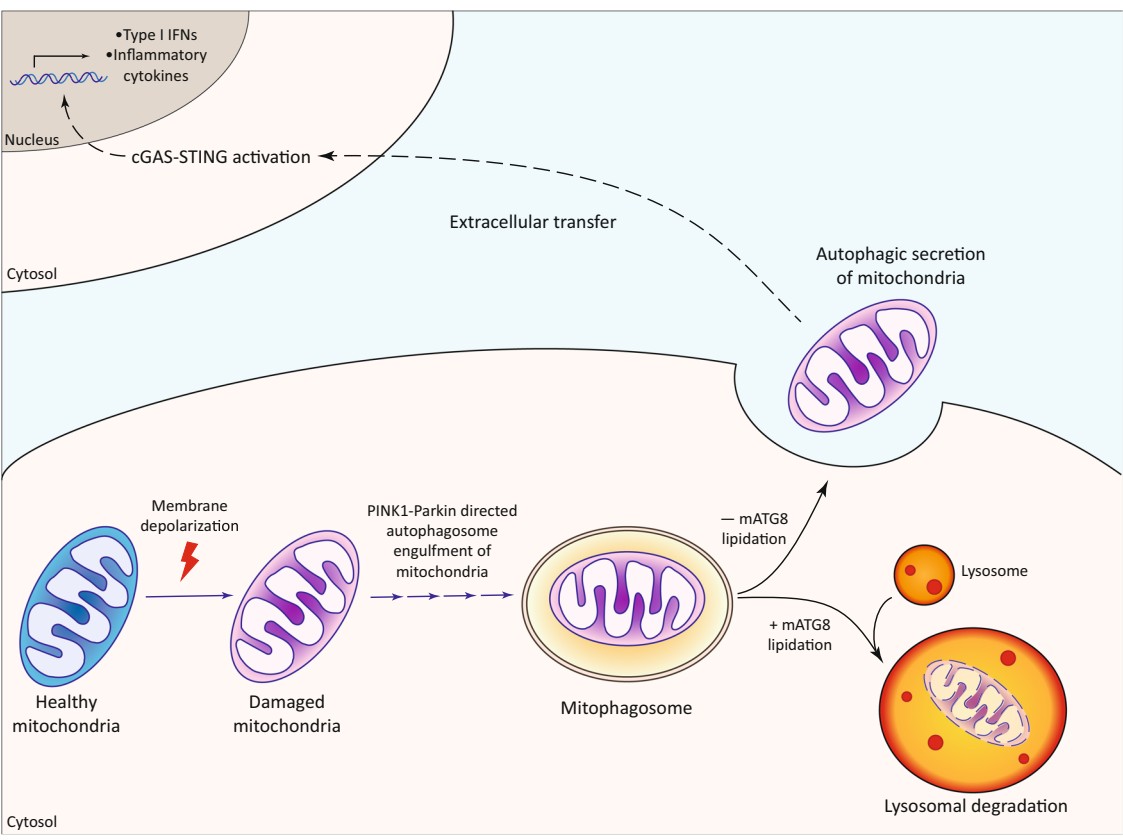

**Fig. 8 Graphic summary of Autophagic Secretion of Mitochondria (ASM).** Damaged mitochondria are cleared from cells via Autophagic Secretion of Mitochondria (ASM) in the absence of mATG8-conjugation. Extracellular mitochondria can activate the cGAS–STING innate immune pathway in recipient cells, thereby promoting an inflammatory response.

may play a role in promoting cGAS–STING-dependent neuroinflammation and contribute to the pathogenesis of neurodegenerative diseases.

In summary, our study presents a MQC pathway via ASM when the mATG8-conjugation system is defective. Given that ASM activates the cGAS–STING-mediated inflammatory response, our study also highlights the importance of the mATG8-conjugation system and therefore mATG8 lipidation in suppressing inflammation, which may have implications in the pathogenesis of inflammatory diseases linked with autophagy defects. Further studies are warranted to dissect the molecular mechanisms that direct autophagosomes towards the plasma membrane for secretion, as well as the pathological implications of ASM in various disease models.

## Methods

**Reagents and antibodies.** Antibodies against citrate synthase [CS] (#14309; 1:1000), GFP (#2956; 1:3000), mitofusin-1 [Mfn1] (#14739; 1:1000), Atg7 (#8558;

1:1000), Atg5 (#12994; 1:1000), Atg3 (#3415; 1:1000), Atg14 (#5504; 1:1000), TBK1/NAK (#38066; 1:1000), PINK1 (#6946; 1:1000), FIP200 (#12436; 1:1000), LC3A (#4599; 1:1000), GABARAP (#13733; 1:1000), GABARAPL1 (#26632; 1:1000), Tom20 (#42406; 1:1000), Aco2 (#6571; 1:1000), SDHA (#11998; 1:1000), CD9 (#13403; 1:1000), Rab7 (#9367; 1:1000), phospho-TBK1/NAK (Ser172) (#5483; 1:1000), phospho-STING (Ser366) (#50907; 1:1000), STING (#13647; 1:1000), phospho-IRF-3 (Ser386) (#37829; 1:1000), IRF-3 (#11904; 1:1000), HRP-linked anti-mouse IgG (#7076; 1:2000), and HRP-linked anti-rabbit IgG (#7074; 1:2000) were purchased from Cell Signalling Technology. Antibodies against MTCO2 (CoxII; ab110258; 1:1000), UQCRC2 (ab203832; 1:3000), Atg9a (ab108338; 1:1000), and NDP52 (ab68588; 1:1000) were purchased from Abcam. Tom70 (sc-390545; 1:1000) and LAMP2 (sc-18822; 1:200) was purchased from Santa Cruz Biotechnology. Anti-LC3B (L7543; 1:3000), anti-tubulin (T5168; 1:5000), and anti-actin (A5441; 1:5000) antibodies were purchased from Sigma-Aldrich. Anti-phospho-ubiquitin (Ser65) (abs1513; 1:1000) and anti-Atg18 (WIPI-2; MABC91; 1:200) antibodies were purchased from Merck Millipore. Anti-DNA (61014; 1:200) antibody was purchased from Progen Biotechnik. Anti-Fumarase (NBP2-59442) and anti-Parkin (H00005071-M01) antibodies were from Novus Biologicals. Goat anti-Mouse IgG (H + L) cross-adsorbed secondary antibody, Alexa Fluor™ 405 (#A-31553; 1:200), goat anti-rabbit IgG (H + L) cross-adsorbed secondary antibody, Alexa Fluor™ 633 (#A-21070; 1:200) and goat anti-mouse IgG

(H + L) cross-adsorbed secondary antibody, Alexa Fluor™ 555 (#A-21422; 1:200) were purchased from ThermoFisher Scientific.

**Chemicals.** Antimycin A (A8674), chloroquine diphosphate salt (C6628) and epoxomicin (E3652) were from Sigma-Aldrich. Oligomycin A (11342), SBI-0206965 (18477), SAR405 (B1286) and Bafilomycin A1 (11038) were from Caymen Chemical. Q-VD-OPh (A1901) was from ApexBio. MG132 (I-130-05M) was from Boston Biochem.

**Plasmids.** pLV-mitoGFP was a kind gift from Pantelis Tsoulfas (Addgene #44385). pMDLg/pRRE (Addgene #12251), pRSV-Rev (Addgene #12253), pMD2.G-VSVg (Addgene #12259) were kind gifts from Didier Trono. pBMN-mCherry-Parkin (Addgene #59419) was a kind gift from Richard Youle. pUMVC-gagpol (Addgene #8449) was a kind gift from Bob Weinberg. pSpCas9(BB)-2A-Puro PX459 (#62988) was a kind gift from Feng Zhang. pFU-Cas9-T2a-miRFP670 was a kind gift from Naiyang Fu (Duke-NUS, Singapore).

**Cell culture.** HeLa and HEK293T cells were cultured in DMEM (HyClone, SH30022.01) supplemented with 10% fetal bovine serum (HyClone, SV30160.03) and maintained at 37 °C in a humidified incubator with 5% $CO_2$. HeLa (CCL-2) and HEK293T (CRL-3216) cells were purchased from American Type Culture Collection (ATCC). Cell lines used were tested and confirmed to be negative for mycoplasma infection. To initiate mitophagy, cells were treated with a combination of 1 μM oligomycin and 1 μM antimycin A in fresh growth media. 20 μM QVD was also added to prevent mitophagy-related cell death. For experiments involving lysosomal inhibition, cells were pretreated with 200 nM BafA1 or 100 μM chloroquine for 2 h before mitophagy induction. For experiments involving proteasomal inhibition, cells were pretreated with 10 μM MG132 or 1 μM epoxomicin for 2 h before mitophagy induction. The ULK1/2 inhibitor SBI-0206965 was added 2 h before mitophagy induction.

**Retroviral/lentiviral generation of stable cell lines.** To generate stably-expressing HeLa cells, lentiviral vectors for mitoGFP (pLV-mitoGFP, pMDLg/pRRE, pRSV-Rev, pMD2.G-VSVg) or retroviral vectors for mCherry-Parkin (pBMN-mCherry-Parkin, pUMVC-gagpol, pMD2.G-VSVg) were transfected into HEK293T-packaging cells using Lipofectamine LTX for 24 h. Fresh media was then added and collected after 36 h. Virus-containing media was added to HeLa cells in the presence of 8 μg/ml polybrene. Positive clones were selected by FACS sorting. HeLa cells stably expressing YFP-Parkin were a kind gift from Dr. Richard Youle.

**Generation of knockout cell lines with CRISPR/Cas9.** To generate knockout lines in YFP-Parkin-expressing HeLa cells, oligonucleotides encoding the guide RNA (gRNA) sequences (listed in Supplementary Data 2) were ligated into *Bbs*I-linearized pSpCas9(BB)-2A-Puro PX459 V2.0 vector. Plasmids were transfected into YFP-Parkin HeLa cells with lipofectamine 3000 and selected with 1 mg/ml puromycin for 4 days. Single clones were then obtained by cell sorting and verified for knockout efficiency by western blotting. For the generation of knockout lines in mCherry-Parkin/mitoGFP expressing cells, cells were first transduced to stably express Cas9 using the lentiviral system (pFU-Cas9-T2a-miRFP670). Cells were subsequently transfected with Alt-R® CRISPR-Cas9 sgRNA containing guide sequences against the gene of interest (listed in Supplementary Data 2). In the case of double-knockout lines, two sgRNAs targeting the two genes of interest were co-transfected into cells. Single clones were subsequently isolated by cell sorting and verified for knockout efficiency by western blotting.

**Animal models and exhaustive exercise.** All animal studies were reviewed and approved by the NUS Institutional Animal Care and Use Committee (IACUC; R17-0195). Muscle specific ATG7 knockout mice were generated by crossing *Atg7*-floxed (*Atg7*f/f) mice with transgenic mice bearing the *Cre* recombinase gene driven by the muscle creatine kinase promoter (Ckmm-cre). *Atg7* flox mice (*Atg7*f/f) were generated and provided by the RIKEN BRC through the National Bio-Resource Project of the MEXT, Japan[87]. *Ckmm-cre* mice were purchased from The Jackson Laboratory (USA)[88]. All mice used in this study were backcrossed to the C57BL/6J background for 7 generations. Mice were housed in a temperature-controlled environment with a 12 h light/dark cycle and had access to food and water ad libitum.

Exhaustive exercise was performed on a treadmill in the Neuroscience Phenotyping Core for 3 consecutive days as previously described[60]. Specifically, 4-month-old male mice were randomly assigned to sedentary (SED) or exhaustive exercise (EE) groups. EE mice were first familiarized with the treadmill for 3 days with low speeds of 8, 10 m/min, and 12/min for 5 min each. Mice were subsequently allowed to rest for 2 days before EE. For EE, the following protocol was used: 10 m/min for 8 min, 15 m/min for 5 min, followed by 1.8 m/min increase every 3 mins till a speed of 22.4 m/min is reached. Then, speed was maintained at 22.4 m/min for 10 mins, followed by an increase of 1 m/min every 5 min until exhaustion. Blood was then harvested via cardiac puncture and allowed to clot for 2 hrs. Clotted blood was spun for 2000 × g for 10 min twice to obtain serum.

**Extracellular vesicle (EV) isolation.** Conditioned media from cells treated as indicated plated onto 150 mm plates were collected and spun down at 300 × g for 10 min to remove dead cells, followed by 2000 × g for 30 min to deplete cell debris. Supernatants were then concentrated with Amicon Ultra-15 centrifugal units (10 kDA cutoff). Concentrates were then ultracentrifuged at 120,000 × g for 2 h (TLS-55 rotor). The EV pellets were then washed in PBS to remove contaminating proteins and ultracentrifuged at 120,000 × g for 2 h. For size-exclusion experiments, the conditioned media was filtered with an 0.22 μm PES syringe filter after the 2000 × g centrifugation step. The filtered media was then concentrated and processed as described above.

For further separation of the EV fraction by Iodixanol buoyant density gradient, the EV pellet was diluted in 1.5 ml 40% Optiprep™ (diluted with 250 mM sucrose, 10 mM Tris–HCl 7.4), overlaid with 1.5 ml 20%, 1 ml 10% and 1 ml 5% Optiprep™. Ten 500 μl fractions were collected after 100,000 × g spin (SW55-ti rotor) for 16 h at 4 °C. Fractions were diluted with 8 volumes of PBS and respun at 120,000 × g for 2 h. Pellets from each fraction was lysed in 1× laemmli sample buffer.

For isolation of EVs from serum, 180 μl of serum was diluted in 600 μl of PBS followed by ultracentrifugation at 120,000 × g for 2 h (TLS-55 rotor). The EV pellets were then resuspended in PBS and ultracentrifuged at 120,000 × g for 2 h.

**Nanoparticle tracking analysis.** For independent characterization of vesicle concentration and size distribution, we used the nanoparticle tracking analysis (NTA) system (Nanosight, NS300; NTA version 3.3). Vesicle concentrations were adjusted to obtain ~50 vesicles in the field of view to achieve optimal counting. All NTA measurements were done with identical system settings for consistency.

**nPLEX measurements.** To confer molecular specificity on the sensor, the fabricated Au surface was first incubated with a mixture of polyethylene glycol (PEG) containing long active (carboxylated) thiol-PEG and short inactive methylated thiol-PEG (Thermo Scientific) (1:3 active: inactive, 10 mM in PBS) for 2 h at room temperature. After washing, the surface was activated through carbodiimide crosslinking, in a mixture of excess NHS/EDC dissolved in MES buffer, and conjugated with specific probes (e.g., antibodies). Excess unbound probes were removed by PBS washing. Conjugated sensors were stored in PBS at 4 °C for subsequent use. All sensor surface modifications were spectrally monitored to ensure uniform functionalization.

To establish the sensor amplification, we incorporated enzymatic growth of insoluble optical deposits for signal enhancement, and optimized the substrate concentration and reaction duration to establish the platform. Specifically, EVs were incubated for 10 min with the CD63-functionalized sensor (BD Biosciences). The bound vesicles were then fixed and permeabilized with paraformaldehyde and Triton X-100. The bound vesicles were then labeled with biotinylated antibodies for 10 min: CD63 (Ancell); HSP70 (Biolegend); Parkin (Novus Biologicals); Fumarase (Novus Biologicals); pUbi (Ser65) (Merck Millipore). As a control experiment, an equivalent amount of biotinylated IgG isotype control antibody (Biolegend) was used on the bound vesicles to determine the amplification efficiency. After washing of unbound antibodies, high sensitivity horseradish peroxidase, conjugated with neutravidin (Thermo Scientific), was allowed to react with the bound vesicles, before the introduction of 3,3′-diaminobenzidine tetra-hydrochloride (Life Technologies) as the reaction substrate. All flow rates for incubation and washing were maintained at 3 and 10 μl min$^{-1}$, respectively.

**SILAC labelling.** For SILAC labelling, cells were maintained in SILAC DMEM (-Arg, -Lys) medium (Thermo Scientific, A33822) containing 10% dialyzed fetal bovine serum supplemented with 42 mg/l $^{13}C_6$$^{15}N_4$ L-arginine and 73 mg/l $^{13}C_6$$^{15}N_2$ L-lysine (Cambridge Isotope) or the corresponding non-labelled amino acids, respectively. Media was also supplemented with 0.5% Insulin-Transferrin-Selenium (Life Technologies, 41400045) to maintain growth factor signalling. Successful SILAC incorporation was verified by in-gel trypsin digestion and MS analysis of heavy cellular samples in parallel to EV samples to ensure an incorporation rate of >98%.

**Mass spectrometry analysis of EVs.** WT and ATG7KO cells were seeded at equal numbers per 150 mm dish and treated as described. After collection of conditioned media, cells were lysed and protein quantifications were quantified as a proxy of cell number. The conditioned media from both samples were then mixed before EV isolation (as described above) to avoid any variation that might arise from sample handling. The ratio of protein concentration from light and heavy cell lysates were used to normalize the volume of conditioned media used for mixing before EV isolation.

EV pellets were lysed in SDS lysis buffer and boiled at 95 °C prior to separation on a 12% NuPAGE Bis–Tris precast gel (Thermo Scientific) for 10 min at 170 V in 1× MOPS buffer, followed by gel fixation using the Colloidal Blue Staining Kit (Thermo Scientific). For in-gel digestion, samples were destained in destaining buffer (25 mM ammonium bicarbonate; 50% ethanol), reduced in 10 mM DTT for 1 h at 56 °C followed by alkylation with 55 mM iodoacetamide (Sigma) for 45 min in the dark. Tryptic digest was performed in 50 mM ammonium bicarbonate buffer with 2 μg trypsin (Promega) at 37 °C overnight. Peptides were desalted on

StageTips and analysed by nanoflow liquid chromatography on an EASY-nLC 1200 system coupled to a Q Exactive HF mass spectrometer (Thermo Fisher Scientific). Peptides were separated on a C18-reversed phase column (25 cm long, 75 μm inner diameter) packed in-house with ReproSil-Pur C18-AQ 1.9 μm resin (Dr Maisch). The column was mounted on an Easy Flex Nano Source and temperature controlled by a column oven (Sonation) at 40 °C. A 215-min gradient from 2% to 40% acetonitrile in 0.5% formic acid at a flow of 225 nl/min was used. Spray voltage was set to 2.2 kV. The Q Exactive HF was operated with a TOP20 MS/MS spectra acquisition method per MS full scan. MS scans were conducted with 60,000 at a maximum injection time of 20 ms and MS/MS scans with 15,000 resolution at a maximum injection time of 75 ms. The raw files were processed with MaxQuant[89] version 1.5.2.8 with preset standard settings for SILAC-labeled samples and the re-quantify option was activated. Carbamidomethylation was set as fixed modification while methionine oxidation and protein N-acetylation were considered as variable modifications. Search results were filtered with a false discovery rate of 0.01. Known contaminants, proteins groups only identified by site, and reverse hits of the MaxQuant results were removed and only proteins were kept that were quantified by SILAC ratios in both 'forward' and 'reverse' samples.

The mass spectrometry data have been deposited to the ProteomeXchange Consortium via PRIDE partner repository[90] with the dataset identifier PXD025356.

**Proteinase K (PK) protection assay**. Cells were treated as indicated, collected by trypsinization, washed once with cold PBS and resuspended in homogenization buffer (220 mM sucrose, 70 mM mannitol, 20 mM HEPES, 1 mM EDTA). Homogenates were passed through a 27 G needle to achieve cell lysis. Lysates were then spun at 500 × g for 5 min twice. Supernatants from each treatment were aliquoted into 3 groups for PK protection assay: (1) Untreated; (2) 50 μg/ml PK; (3) 50 μg/ml PK and 1% Triton-X100 for 15 min on ice. Reaction was terminated by addition of 2 mM PMSF to all tubes. TCA precipitation was then performed, and the protein pellets were resuspended in 1× Laemmli sample buffer and boiled for 10 min.

For the PK protection assay, EV pellets isolated by differential ultracentrifugation as described above were resuspended in PBS and split into three groups: (1) Untreated; (2) 50 μg/ml PK; (3) 50 μg/ml PK and 1% Triton-X100 for 30 min on ice. The reaction was terminated by addition of 2 mM PMSF. 4× Laemmli sample buffer was added directly to the samples and boiled for 10 min.

**Immunoblotting**. Cells or EV pellets were homogenized in SDS lysis buffer (62.5 mM Tris–HCl, pH 6.8, 25% glycerol, 2% SDS, 1 mM dithiothreitol, 1× phosphatase and proteinase inhibitor cocktail [Thermo Fisher Scientific, 78446). Lysates were boiled for 10 min and centrifuged at 13,000 × g for 10 min to remove insoluble cell debris. The supernatant was diluted in Laemmli sample buffer and equal amounts of proteins were separated in 10% or 6–12% gradient SDS–PAGE gels, followed by transfer to PVDF membranes (Bio-Rad Laboratories, 1620177). Membranes were blocked with 5% non-fat milk, incubated overnight with primary antibodies, washed, and then incubated with secondary antibodies. Proteins were then visualized by chemiluminescence with the ImageQuant LAS 500 (GE Healthcare). Immunoblots were quantified with FIJI v1.53 (ImageJ).

**Immunofluorescence**. Cells were seeded on 12 mm glass coverslips, left to attach overnight and treated as indicated. Cells were fixed in 4% paraformaldehyde for 20 min, washed three times in PBS, permeabilized in 0.1% Triton X-100 in PBS for 5 min and blocked in blocking buffer (5% FBS & 2% BSA in PBS) for 1 h. Cells were then incubated in primary antibodies overnight, followed by three washes in PBS and incubation in secondary antibodies for 1.5 h. Samples were then mounted in ProLong™ Diamond anti-fade mounting medium with DAPI (Life Technologies, P36962), and imaged on an Zeiss LSM 710 confocal microscope at ×60 magnification using ZEN Black 2011 (Zeiss) or Olympus FV3000 confocal microscope at ×100 magnification using FV31-ASW (Olympus). For mtDNA immunostaining, 10 z-stack slices encompassing the entire cell was taken per field. Analysis of mtDNA volume was performed using Imaris v7.4 (Oxford Instruments) as previously described[12]. Specifically, cells were individually processed with image segmentation using surface smoothing of 90 nm, background subtraction of 300 nm, and an intensity threshold of 100. Identified structures were identified as mtDNA volumes if the DAPI channel intensity did not exceed the value of 1200.

**Transmission electron microscopy**. For electron microscopy (EM) studies, cells were fixed with EM fixation buffer (2.5% glutaraldehyde, 0.1 M phosphate buffer pH 7.4), followed by 1% OsO4. The cells were further dehydrated, cut into thin sections and stained with uranyl acetate and lead citrate. All the images were obtained using a JEM 1016CX electron microscope with a digital camera.

**Quantitative real time PCR**. Total RNA was extracted from cells using the PureLink™ RNA Mini Kit (Life Technologies) following the manufacturer's protocol. cDNA was prepared using iScript™ cDNA Synthesis Kit (Bio-Rad, 1708890) for RT-PCR. Quantitative real-time PCR (qRT-PCR) reaction consisted of 1 × iTaq™ Universal SYBR® Green Supermix (Bio-Rad, 1725120), 10 μM each of gene-specific forward and reverse primers, and 30 ng cDNA per reaction. qRT-PCR was performed in a 96-well plate format, and reactions were carried out on the Bio-Rad CFX96 real-time PCR System using CFX Mananger (Biorad). The abundance of target gene mRNA was normalized to housekeeping gene *TBP*. Primer sequences are listed in Supplementary Table 2.

**LDH release assay**. LDH release from cells were measured using the CyQUANT™ LDH Cytotoxicity Assay (Thermo Fisher, C20301) following the manufacturer's instructions.

**Statistics and reproducibility**. Data are presented as the mean ± SEM or stated otherwise. P values were calculated using Prism 8.0 (GraphPad) for two-way ANOVA or Microsoft Excel 2016 (Microsoft) for Student's T-test. Statistical significance was accepted at $P < 0.05$. Graphs were generated using Prism 8.0 (GraphPad) or OriginPro 2019b (OriginLab). Graphic images were drawn using Abode Illustrator v26.3.1 (Abode). Experiments were replicated four times in Fig. 1a, c, Supplementary Fig. 1a, b; replicated thrice for Figs. 2a–f, 5a, 6a–e, 7a, c–h; and replicated twice for Figs. 4a, d–f, 5d–h, Supplementary Figs. 2a, g–I, and 3a.

**IL-6 ELISA**. Conditioned media from recipient cells treated with EVs as indicated were collected after 24 h. Media was spun at 500 × g for 10 min to clear cell debris before measurement of IL-6 concentration using the human IL-6 ELISA kit (Invitrogen, BMS213-2) as per manufacturer's instructions.

**Reporting summary**. Further information on research design is available in the Nature Research Reporting Summary linked to this article.

## Data availability

Data supporting this study are provided within the paper and supplementary files. The mass spectrometry data generated from this study has been deposited to the ProteomeXchange Consortium via PRIDE partner repository under the accession code PXD025356. Source data are provided with this paper.

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

## Acknowledgements

We thank Naiyang Fu from Duke-NUS for providing the pFU-Cas9 plasmid and advice on CRISPR-Cas9 genome editing. We also thank members of the Shen lab for the discussion and input. We thank Liou Yih Cherng and Deng Lih Wen from NUS for their valuable advice and support of this work. We also thank the Neuroscience Phenotyping Core (NUS) for their help with treadmill exercise experiments. This study was supported by research grants from Singapore National Medical Research Council (NMRC/CIRG/1490/2018) and Ministry of Education grant (MOE2018-T2-1-060), Macau Science and Technology Development Fund (FDCT0078/2020/A2 and 0031/2021/A1) to H.M.S.; Singapore National Research Foundation to H.M.S. and S.C.L. (NRF2019-NRF-ISF003-3221); and Singapore National Medical Research Council (OFIRG18nov-0093) and Ministry of Education grant (NUHSRO/2018/012/T1) to S.Y.T. H.W.S.T., Y.L.C., H.D., and A.N. are supported by NUS Graduate School Research Scholarships.

## Author contributions

H.W.S.T. and H.M.S. conceptualized the project. H.W.S.T., G.L., Y.L.C., L.M.W., R.T., and H.M.S. designed and performed the experiments and analyzed the data. A.N. and H.S. performed the NTA and nPLEX experiments. C.C. and D.K. performed the proteomics experiments. H.W.S.T., S.C.L., H.D. and S.Y.T. were involved in the animal experiments. W.X.D. performed the EM analysis. H.W.S.T. and H.M.S. wrote the manuscript. All authors reviewed and edited the manuscript.

## Competing interests

The authors declare no competing interests.
