## [Peer Review File · Nature Communications]

A degradative to secretory autophagy switch mediates mitochondria clearance in the absence of the mATG8-conjugation machineryREVIEWER COMMENTS

Reviewer #1 (Remarks to the Author):

The manuscript by Tan et al., describes an alternative mitochondria clearance pathway, related to the secretory autophagy pathway. By gene deletion of different players of the ATG8 pathway, authors demonstrate that the mATG8 lipidation machinery is not required, and that its absence switches the clearance pathway from the lysosomal fusion and degradation to the secretion of the mitochondria to the extracellular media.

However, the nature of the secreted material is not explored in depth. Authors isolate mitochondrial components by sequential ultracentrifugation, but they claim that they are not included in exosomes because they are removed by filtration through a 0,22 micrometre filter.

It is thus not clear from their data the exact nature of the mitochondria-containing particles. The term EV they use for the ultracentrifugation pellet is a very general term that includes everything secreted by a cell that is surrounded by a membrane. Besides, filtration by 0,22 is not a standardized procedure to extract exosomes from an ultracentrifugation pellet.

Mitochondria have already been shown to be secreted in several different ways: as exospheres (Melentijevic et al., 2017; Nicolás Avila et al., 2020), double-membrane mitovesicles (d'Acunzo et al., 2021) or CD63+ EVs (Peruzzotti-Jametti et al., 2021; Suarez et al., 2021). In some cases these mitochondrial-containing vesicles retain respiratory functionality and can be isolated from classical EVs (d'Acunzo) or are present in endosome-derived vesicles, even if filtration reduces the presence of mitochondrial derived components from the pellet (Peruzzotti-Jametti). Thus, the data here provided is not conclusive to suggest that mitochondrial components are not in exosomes and authors should explore in depth the nature of the mitochondrial-containing structures.

The fact that mitochondrial-secreted components can induce a proinflammatory phenotype has also been repeatedly described (Peruzzotti-Jametti, Torralba et al., 2018 among others).

Reviewer #2 (Remarks to the Author):

In this manuscript, Tan et al. investigate the role of the autophagy conjugation machinery, particularly ATG7, in the clearance of damaged mitochondria. Using CRISPR KO of ATG7 in HeLa cells, they induce mitophagy using antimycin A and oligomycin, and examine the effects on select outer mitochondrial membrane proteins, inner mitochondrial membrane proteins and matrix proteins over time. Unexpectedly, they found that levels of mitochondrial proteins were still reduced in the ATG7KO cells similar to control. KOs of other conjugation machinery components (ATG5, ATG3) were used to confirm

this finding, though these experiments all lacked quantitation. The authors then examine KOs of upstream autophagosome biogenesis components (ATG9A, FIP200, ULK1 chemical inhibitor) and conclude, in contrast, that they are required for a decrease in cellular levels of mitochondrial proteins. The degradation and/or clearance of mitochondria was found to be related to extracellular release, and the authors rule out small extracellular vesicles as the release mechanism. Instead, they conclude that mitochondrial clearance occurs by secretory autophagy, and call this process Autophagic Secretion of Mitochondria (ASM). This conclusion is supported by TEM analysis and by double KOs with the autophagosome biogenesis components vs the lysosomal fusion protein RAB7A. Lastly, the authors isolate secreted particles from ATG7KO cells and explore their functional effects on activation of cGAS-STING components, previously shown to be induced by mtDNA. Previous studies, referenced by the authors, have reported the extracellular release of mitochondria. In this study, the proposed mechanism of secretory autophagy for release of damaged mitochondria (ASM) is novel and has important disease implications – while some aspects of this model are supported, other aspects require replication/quantitation or further validation to convincingly support the authors claims as detailed below.

Concerns:

1. Figure 1a and 1b: The authors indicate mitochondrial degradation occurs at a “slower rate” in ATG7, ATG5 and ATG3 KO lines vs WT. The differences appear subtle and there is a lack of quantitation to back up this statement. Replicate blots with quantitation are required.

2. Figure 2: Quantitation of replicate blots and corresponding statistical analyses are required to back up the authors conclusion that ATG components involved in autophagosome biogenesis upstream of the ATG8 conjugation system are required for mito clearance. For example, in 2a, the stated difference in mitoGFP and COXII levels between PINK1KO and ATG7KO are not apparent. In Figure 2c, the differences between ATG9A KO and ATG7KO are subtle, with the exception of CS, and do not adequately support the claim that “degradation in ATG9A was completely blocked”. In Fig 2D, while the FIP200 KO data shown does suggest impaired degradation relative to WT, the pattern in the FIP200KO very closely resembles the pattern shown in the ATG7KO – yet the conclusions regarding degradation are opposite.

3. Supplementary Figure 2b: the effects of BafA1 and CQ are subtle and also require quantitation (with replicates and statistics).

4. Figure 4: Panel 4c depicts nanoparticle analysis from ATG5KO cells, but the EV proteome analysis (Fig 3), EV western analyses and serum EV analyses were all from ATG7KO. For consistency the nanoparticle tracking analysis should be performed for ATG7KO cells.

5. Figure 5: Panel 5a: The statements regarding altered levels of NDP52 require replicates and quantitation.

Panels c-e nicely support the proposed role for autophagosome formation vs lysosomal fusion (panel f). However, to support the conclusion regarding a secretory autophagy route, this figure should also include a similar panel with KO of a distal (ie. plasma membrane) component of secretory autophagy such as STX3, STX4, or SNAP29. Currently, the authors cannot rule out a novel mechanism of secretion. Also, the authors have not convincingly excluded a role for ATG5 in ASM (based on the single western blot in Fig 1b) as stated in the Discussion on p. 13.

6. Figure 6: The western blots shown do not convincingly support the conclusions made. In panels a and b, inhibition of mitochondrial protein degradation in the DKO is variable (ie. protein dependent), appearing similar to the ATG7KO for some proteins. In particular, the comparison of panel d and e is concerning since the western blots appear highly similar yet the authors conclude that degradation was “almost completely blocked” (in d) versus “no effect on mitochondrial degradation” (in e).

7. Figure 7a: Quantitation, replicates and statistical analyses are required to back up the statement of “significantly higher levels of cGAS-STING activation”. While STING-dependent genes appear to be altered, these effects are not necessarily attributable to released mitochondrial components (mitochondrial DNA) as suggested.

8. Supplementary Fig4: WIPI2 puncta require quantitation to support the conclusion made in the text that they were “greatly reduced”.

9. The isolated vesicles were not purified nor characterized sufficiently to be called exosomes. Throughout the text, extracellular or secreted “particles” (or small particles in the case of filtration) would be a better term since secreted material was not characterized as vesicular and likely contains many different components. This would also help to avoid confusion and improve consistency with the working model in Figure 8.

10. The authors refer to “mATG8 lipidation independence” throughout the text, but the majority of experiments were performed in ATG7KO (or DKO) backgrounds only, and assessed LC3B lipidation only. These statements should be corrected to “ATG7 independence” unless supported by appropriate additional lines of evidence.

11. Methods indicate that cells were treated with 100uM of CQ. This concentration seems very high as levels used are typically up to 10uM.

Reviewer #3 (Remarks to the Author):

In this manuscript Tan et al. employ an impressive number of cell biological, biochemical and proteomics approaches as well as animal models to report the surprising finding that the clearance of damaged mitochondria occurs via secretion in cells lacking the ability to lipidate ATG8 family proteins. One of the main pathways for the removal of damaged mitochondria is termed PINK1/Parkin mediated mitophagy. In this pathway the damaged mitochondria are engulfed by autophagosomes. Canonically, this pathway entails the attachment of the ubiquitin-like ATG8 family proteins to membrane lipids residing in the nascent autophagosomal membrane. Here the authors delete key components of the ATG8 lipidation machinery (ATG7, ATG3, ATG5) and observe that damaged mitochondria can still be engulfed by autophagosomes, as observed before (PMID: 27864321). In addition, they make the unexpected observation that rather than accumulating autophagosomes containing mitochondria, these cells secrete them into the extracellular space in pathway the authors call ASM. A shortcoming of the study is that it remains unclear if ASM occurs also when mATG8 are present and can be lipidated? Is this an artificial pathway only observed in KO cells or a pathway that always occurs at low levels and is upregulated when the mATG8s can't be conjugated? Some evidence for the latter would make this very comprehensive study of interest for a much wider research community.

Major comments:

1. To be absolutely sure that mATG8 lipidation is abolished in the ATG7, ATG3 and ATG5 KO cells the authors should blot for more mATG8 family members apart from LC3B, in particular as a faint band that runs close to the lipidated form of LC3B can be observed for these KO cell lines in Figure S1.
2. Figure 2b: The TBK1 KO does not seem to be complete. How was the KO of TBK1 verified? How can the authors therefore be sure that TBK1 is not required for mitochondrial clearance? Same for ATG9A (Figure 2c), which also doesn't seem to be completely depleted.
3. There seems to be significant loss of mitochondrial IMM proteins in FIP200 KO cells. In order to fully appreciate the relative importance of the various autophagy factors the authors have analyzed regarding the clearance of mitochondria it would be important to quantify the blots.

4. Figure 2f: the authors should provide evidence that their BafA1 treatment effectively inhibits lysosomal degradation, for example by assessing p62 accumulation. In addition, can the authors please elaborate on why inhibition of proteasomal activity blocks clearance of all mitochondrial proteins.

5. Figs 3 and 4: The authors should determine the cell viability of the ATG7KO vs WT cells after A/O treatment to rule out that the increase of extracellular proteins including mitochondrial proteins is due to cell death associated cell lysis.

6. Fig S3a, page 8 bottom: the authors write that they observe “single-membrane vesicular structures that resembled mitochondria with ruptured outer membranes”. This statement is confusing to this reviewer. If the mitochondria are released by secreted autophagosome then the outer autophagosomal membrane fuses with the plasma membrane leaving behind a single membrane vesicle (the former inner autophagosomal membrane) containing the damaged mitochondrion. Thus, in principle three membranes should be observed. Is this what the authors see? Unfortunately, the data shown in Fig S3 do not seem to support this.

7. Figure 6: the data should be quantified and directly compared. The block in IMM protein clearance by ATG14, FIP200 and ATG9A KO appears to be rather minor.

8. Figure 7: Did the authors correct for the higher number of EVs in ATG7 KO cells? Is it possible that the increased activation of the cGAS-STING pathway by the ATG7 KO derived EVs is due to a higher number of EVs? In this respect, it would be advisable to include an additional control cell line in which PINK1 is deleted on top of ATG7 in order to test if this activation is indeed due to secreted mitochondria.

9. From Figure 3 on the authors employ solely ATG7 KO cells, which have defects in ATG8 lipidation but also fail to form the ATG12–ATG5 conjugate. This conjugate in turn may have functions that are not related to ATG8 lipidation. The authors should point out this complication more clearly in the manuscript.

Minor comments:

10. The authors should consider quantifying their key data derived from western blots.

11. Fig 3d: The legend should state more clearly what exactly is shown. Are these the GO analyses of the proteins enriched in the EVs of ATG7 KO vs WT cells?

12. Page 10: The authors write that they used ATG9A as a cytosolic control. ATG9A is a transmembrane protein and as such unlikely to be cytosolic. In fact, the protein has been shown to localize to small Golgi derived vesicles. This sentence should be rephrased.

Reviewer #4 (Remarks to the Author):

Mitochondrial quality control (MQC) is key to the maintenance of mitochondrial homeostasis, and is critically involved in many pathophysiological processes. This manuscript identifies autophagic secretion of mitochondria (ASM) as a novel MQC mechanism through the clearance of damaged mitochondria in an ATG8 (LC3) lipidation-independent manner. The authors used quantitative proteomic analyses and profiled the protein content of the extracellular vesicles (EVs) derived from cells undergoing mitochondrial stress. In doing so, they were able to show that the damaged mitochondria were secreted via ASM, rather than the exosome-dependent pathway. Finally, they found that the ASM-derived mitochondria promoted innate immune signaling in a cGAS-STING-dependent manner. Although previous studies demonstrated that damaged mitochondria could be extruded from cells into extracellular vesicles, the current study is potentially interesting and provides a novel perspective and role of autophagic secretion in this process. Generally, the experiments shown are of good quality. The major weaknesses of the paper concern several mechanistic aspects of ASM and its physiological relevance. Certain experimental results were inconsistent with respect to the claims, and there were several key controls that were missing.

Specific points are shown below:

1. The molecular mechanism of autophagosome-mitochondria fusion is unclear. When the lipidation of ATG8 (LC3) is blocked, the authors found that damaged mitochondria were sequestered by intact autophagosomes (or mitophagosomes), and were secreted to the extracellular space. However, previous studies have shown that the fusion of autophagosomes with mitochondria is mediated by lipidated LC3 with the adaptor proteins (e.g., P62, OPTN and NDP52) or mitochondrial proteins (e.g., BNIP3, NIX and FUNDC1) (Nature Cell Biology, volume 20, pages 1013–1022 (2018)). The authors should explain which protein(s) might mediate this mitochondria-autophagosome fusion, besides ATG8 (LC3).
2. The biological relevance of ASM is not fully understood. In the EE model, which tissues/organs represent the source of the extracellular mitochondria? Related to the findings that ASM-derived mitochondria activate cGAS-STING signaling, did the authors observe any signs of inflammation in the EE mice? The authors should also add additional experiments demonstrating the physiological relevance of ASM, e.g., using the cardiomyocyte hypoxia model (<https://doi.org/10.1016/j.cmet.2021.08.002>).

3. Some key controls were missing. As an example, for the SILAC experiment (Fig. 3A), the authors should add the wild type and ATG7 KO cells (without the A/O treatment) as the controls. Because only low amounts of extracellular vesicles were derived from these samples, the authors should explain how they performed the normalization procedures (at the sample preparation step and during bioinformatic analyses). The authors should also explain how they removed the secreted proteins (i.e., those derived from the conditional media) from the isolated extracellular vesicles. Finally, the authors should describe how they differentiated genuine secreted proteins from those that were non-specifically released from dead cells.

4. In various panels of Figure 5 and Figure 6, the authors should also include WT cells as the controls.

5. The authors should stain the mtDNA in the EVs or track the secretion of mitochondria using mtDNA staining.

6. The authors should measure the cell viability and membrane integrity of the A/O-treated cells (in particular for the 24 hr-treatment group). This is an important consideration because there is a possibility that mtDNA was released, owing to cell death, rather than through the EVs.

7. Page 11. The authors should elaborate the mechanism by which ASM induces the activation of cGAS-STING signaling in the recipient cells. The mtDNA is encapsulated in the EVs, and is not in direct contact with cGAS in the recipient cells. This is a potential gap in the model and the authors need to provide more evidence to substantiate their hypotheses.

Other comments:

1. In several of the Figures (e.g., Fig 1A, 1B and 2A), there were two bands for mitoGFP (in particular in cells treated with A/O). The authors should explain this.

2. Fig. 1C, author should perform the LC3 staining and lysosome (LAMP1 or LAMP2) staining to check the co-localization of mtDNA with the autophagosomes and lysosomes.

3. In several of the Figures (e.g., Fig S1A and S1B), the LC3I/LC3II levels were not changed upon the treatment of A/O. These results are inconsistent with the previous report (<https://doi.org/10.1016/j.cell.2016.11.042>). Furthermore, it seems that the LC3II level was very low in the control conditions. The authors need to discuss these results.

4. Fig. 2C, It seems that ATG9A shows up as a smear. The authors should clearly label which corresponds to the ATG9A band.

5. The authors claimed that “The degradation of mitochondria markers was completely blocked in ATG9A-KO cells as compared to WT or ATG7-KO cells (Fig. 2c). Additionally, the knockout of FIP200 in cells impaired the degradation of mitochondrial proteins (Fig. 2d). However, in Fig. 2C and 2D, COXII and UQCRC2 were degraded even in the ATG9A KO or FIP200KO cells upon the 24-hr treatment of A/O. These results were inconsistent with the authors’ conclusion.

6. The authors claimed that “Inhibition of lysosomal activity with lysosome inhibitors bafilomycin A1 (BafA1) or chloroquine (CQ) resulted in the partial impairment of IMM and matrix protein degradation induced by A/O, while the degradation of OMM proteins were unaffected.” However, in Fig. 2F and Fig. S2d, the degradation of CS, COXII and UQCRC2 was partly blocked by BafA1 in ATG7KO and ATG5/ATG3-KO cells. The authors need to explain these observations.

7. Fig. 3c and 3d, many of the identified proteins were derived from the lysosomes (e.g., mTOR, LAMTOR1, LAMP1 and LAMP2). The authors should address the possibility that the damaged mitochondria might fuse with the autophagosome-lysosomes?

8. Fig. 3, the authors did not identify ACO2, COXII, Tom70, Tom20, MFN1, CD9 or CD63 in the MS-based proteomic experiments. These results were inconsistent with the authors’ conclusion that these proteins were increased in the EVs from ATG7 KO cells.

9. Fig. 4A, the level of CD63 was lower in both WCL and EVs from the ATG7 KO cells without A/O treatment, compared to the WT cells. The authors should repeat these experiments and check the expression of CD63.

10. Fig. 4b, the level of LC3 was not changed upon EE in the WT mice. The authors need to explain this observation.

11. Fig. 5A, the authors should explain why there was less protection of NDP52 against proteinase K digestion in ATG7KO cells compared to that in WT cells. Based on the hypothesis proposed by the authors, NDP52 should be equally protected against proteinase K digestion under these two conditions.

12. The pattern of ACO2 in Figure 5d is different from that in the other Figures (e.g., Figure 5C, 5E and 5F). The authors should repeat these western blotting experiments.

13. Fig. 6, the authors should add the WT cells as the control. The authors claimed that “When compared to ATG7-KO cells, the degradation of IMM and matrix proteins were significantly inhibited in ATG7/ATG14-DKO or ATG7/FIP200-DKO cells upon A/O treatments (Fig. 6a-6b). However, it seems that the degradation of matrix and IMM proteins were not dramatically affected in the ATG7/FIP200-DKO cells upon 24-hrs of A/O treatment. The authors should clarify this.

14. Fig. 7, the authors should measure and stain the mtDNA in the EVs. The authors need to demonstrate how mtDNA is released from the EVs to activate cGAS-STING in the recipient cells.

15. Fig.7C, the authors should exclude the possibility that a portion of IL6 is derived from the EVs during the isolation step.

REVIEWER COMMENTS

Reviewer #1 (Remarks to the Author):

The manuscript by Tan et al., describes an alternative mitochondria clearance pathway, related to the secretory autophagy pathway. By gene deletion of different players of the ATG8 pathway, authors demonstrate that the mATG8 lipidation machinery is not required, and that its absence switches the clearance pathway from the lysosomal fusion and degradation to the secretion of the mitochondria to the extracellular media.

However, the nature of the secreted material is not explored in depth. Authors isolate mitochondrial components by sequential ultracentrifugation, but they claim that they are not included in exosomes because they are removed by filtration through a 0,22 micrometre filter. It is thus not clear from their data the exact nature of the mitochondria-containing particles. The term EV they use for the ultracentrifugation pellet is a very general term that includes everything secreted by a cell that is surrounded by a membrane. Besides, filtration by 0,22 is not a standardized procedure to extract exosomes from an ultracentrifugation pellet. Mitochondria have already been shown to be secreted in several different ways: as exospheres (Melentijevic et al., 2017; Nicolás Avila et al., 2020), double-membrane mitovesicles (d'Acunzo et al., 2021) or CD63+ EVs (Peruzzotti-Jametti et al., 2021; Suarez et al., 2021). In some cases these mitochondrial-containing vesicles retain respiratory functionality and can be isolated from classical EVs (d'Acunzo) or are present in endosome-derived vesicles, even if filtration reduces the presence of mitochondrial derived components from the pellet (Peruzzotti-Jametti). Thus, the data here provided is not conclusive to suggest that mitochondrial components are not in exosomes and authors should explore in depth the nature of the mitochondrial-containing structures.

Response: We thank the Reviewer for the critical and insightful comments. Here we would like to argue that the goal of the filtration experiment was not to remove exosomes from the EV preparation. Instead, filtration was performed to remove the secreted mitochondria from the EV preparation, thereby showing the secreted mitochondria is a distinct population from small EVs including endosome or plasma membrane-derived vesicles. This method was also utilized by Peruzzotti-Jametti et al (Peruzzotti-Jametti et al., 2021) as mentioned by the Reviewer.

Following the Reviewer's comments, we have now revised our Results section to explain the rationale of this experiment more clearly (Page 9, line 244-248). Additionally, we acknowledge that the paradigm in the EV field is rapidly evolving, as both plasma membrane or endosome-derived small EVs have been shown to share protein markers and flotation densities (Kowal et al., 2016; Mathieu et al., 2021). As such, we have decided to replace the term "exosomes" for a more general term "small EVs" whenever appropriate throughout our manuscript.

Additionally, we do agree with the Reviewer that the 0.22 um filtration of the conditioned media alone is insufficient to prove that the secreted mitochondria are not found in small

EVs. To address this potential issue, our study was complemented by the following experiments:

First, we employed a proteinase K protection assay (**Figure 4e**) which is a well-established method to determine if cargos are loaded within EVs (Bonsergent and Lavieu, 2019). We showed that mitochondrial proteins were not protected from proteinase K digestion unlike the EV cargo HSP70, suggesting that mitochondria are unlikely to be inside small EVs.

Second, with support from our collaborators, we used a nPLEX sensor (**Suppl Figure 3e**) which is a microfluidic device designed to immunocapture and quantify CD63+ EVs, as well as to molecularly profile their cargos (Im et al., 2014). Profiling of CD63+ EVs by nPLEX revealed no changes in the mitochondria markers among the different treatment groups, further suggesting that mitochondria were not found within CD63+ EVs.

Third, we have performed a new bottom-up iodixanol density gradient experiment to separate different EV populations and our new data showed that the mitochondrial markers did not co-fractionate with the EV marker CD9, further confirming that the secreted mitochondria and small EVs were distinct populations. The new data have been presented in **Figure 4f**. We have revised the Results section accordingly (**Page 9-10, line 251-255**).

We have also updated our references to include the work of Nicolás Avila et al. (2020), d'Acunzo et al. (2021) and Peruzzotti-Jametti et al. (2021) in the Introduction section (**Page 4, line 89-92**).

The fact that mitochondrial-secreted components can induce a proinflammatory phenotype has also been repeatedly described (Peruzzotti-Jametti, Torralba et al., 2018 among others).

Response: We thank the Reviewer for raising this relevant point. We agree with the Reviewer that it has been previously reported by multiple groups that secreted mitochondria can induce a proinflammatory phenotype (Joshi et al., 2019; Peruzzotti-Jametti et al., 2021; Torralba et al., 2018). However, our work further establishes a link between defective autophagy and enhanced proinflammatory response through the secretion of mitochondria, which would provide a novel mechanistic explanation for inflammation in multiple autophagy-related diseases (especially neurodegenerative and autoimmune disorders with defective autophagy) (Levine and Kroemer, 2019). We feel that this is of importance, given the growing appreciation of inflammation in the etiology of these diseases (Furman et al., 2019; Kwon and Koh, 2020).

Reviewer #2 (Remarks to the Author):

In this manuscript, Tan et al. investigate the role of the autophagy conjugation machinery, particularly ATG7, in the clearance of damaged mitochondria. Using CRISPR KO of ATG7 in HeLa cells, they induce mitophagy using antimycin A and oligomycin, and examine the effects on select outer mitochondrial membrane proteins, inner mitochondrial membrane proteins and matrix proteins over time. Unexpectedly, they found that levels of mitochondrial proteins were still reduced in the ATG7KO cells similar to control. KOs of other conjugation machinery components (ATG5, ATG3) were used to confirm this finding, though these experiments all lacked quantitation. The authors then examine KOs of upstream autophagosome biogenesis components (ATG9A, FIP200, ULK1 chemical inhibitor) and conclude, in contrast, that they are required for a decrease in cellular levels of mitochondrial proteins. The degradation and/or clearance of mitochondria was found to be related to extracellular release, and the authors rule out small extracellular vesicles as the release mechanism. Instead, they conclude that mitochondrial clearance occurs by secretory autophagy, and call this process Autophagic Secretion of Mitochondria (ASM). This conclusion is supported by TEM analysis and by double KOs with the autophagosome biogenesis components vs the lysosomal fusion protein RAB7A. Lastly, the authors isolate secreted particles from ATG7KO cells and explore their functional effects on activation of cGAS-STING components, previously shown to be induced by mtDNA. Previous studies, referenced by the authors, have reported the extracellular release of mitochondria. In this study, the proposed mechanism of secretory autophagy for release of damaged mitochondria (ASM) is novel and has important disease implications – while some aspects of this model are supported, other aspects require replication/quantitation or further validation to convincingly support the authors claims as detailed below.

Response: We thank the Reviewer for the positive feedback and critical comments. We have revised our manuscript accordingly, as highlighted in detail below.

Concerns:

1. Figure 1a and 1b: The authors indicate mitochondrial degradation occurs at a “slower rate” in ATG7, ATG5 and ATG3 KO lines vs WT. The differences appear subtle and there is a lack of quantitation to back up this statement. Replicate blots with quantitation are required.

Response: We thank the Reviewer for the suggestion. Following the advice, we have now provided immunoblot quantifications for **Figure 1a and 1c** in **Figure 1b and 1d**, respectively. While we did observe that mitochondrial degradation was slightly delayed at the 12 hr time point, this effect was indeed subtle as mentioned. As such, we have decided to remove the statement “slower rate” in this revised manuscript.

2. Figure 2: Quantitation of replicate blots and corresponding statistical analyses are required to back up the authors conclusion that ATG components involved in autophagosome biogenesis upstream of the ATG8 conjugation system are required for mito

clearance. For example, in 2a, the stated difference in mitoGFP and COXII levels between PINK1KO and ATG7KO are not apparent. In Figure 2c, the differences between ATG9A KO and ATG7KO are subtle, with the exception of CS, and do not adequately support the claim that “degradation in ATG9A was completely blocked”. In Fig 2D, while the FIP200 KO data shown does suggest impaired degradation relative to WT, the pattern in the FIP200KO very closely resembles the pattern shown in the ATG7KO – yet the conclusions regarding degradation are opposite.

Response: We thank the Reviewer for the critical comment. We have now provided immunoblot quantifications for **Figure 2a to 2d** in **Suppl Figure 2b to 2e**, respectively, to support our conclusion. We have also changed the term “completely blocked” to “significantly inhibited” for **Figure 2c** to tone down the claim. We have revised the Results section accordingly (**Page 7, line 162**).

Moreover, as we are unsure why our previous FIP200 knockouts showed a significant degradation of CS specifically, we have generated new FIP200 KO cells by using a dual sgRNA design as previously described (Vargas et al., 2019) and performed new experiments with this new FIP200 KO cells. The new data are now presented in **Figure 2d** with the corresponding quantification data in **Suppl Figure 2e**.

3. Supplementary Figure 2b: the effects of BafA1 and CQ are subtle and also require quantitation (with replicates and statistics).

Response: We thank the Reviewer for the question. A similar request was also made by Reviewer #4. We have provided quantifications for **Figure 2f** in **Suppl Figure 2f**, showing a partial inhibition of mitochondrial degradation by BafA1 as described in our Results section (**Page 7, line 172-174**).

4. Figure 4: Panel 4c depicts nanoparticle analysis from ATG5KO cells, but the EV proteome analysis (Fig 3), EV western analyses and serum EV analyses were all from ATG7KO. For consistency the nanoparticle tracking analysis should be performed for ATG7KO cells.

Response: We thank the Reviewer for the suggestion. Following the advice, we have now replaced **Figure 4c** with a new nanoparticle tracking analysis (NTA) experiment from ATG7 KO cells and revised the Results section accordingly (**Page 9, line 235-236**). The NTA analysis data from ATG5-KO cells have been moved to **Suppl Figure 3d**.

5. Figure 5: Panel 5a: The statements regarding altered levels of NDP52 require replicates and quantitation.

Response: Following the advice, we have provided quantifications in **Figure 5b** for the percentage of Proteinase K-protected NDP52 shown in **Figure 5a**.

Panels c-e nicely support the proposed role for autophagosome formation vs lysosomal fusion (panel f). However, to support the conclusion regarding a secretory autophagy route, this figure should also include a similar panel with KO of a distal (ie. plasma membrane) component of secretory autophagy such as STX3, STX4, or SNAP29. Currently, the authors

cannot rule out a novel mechanism of secretion. Also, the authors have not convincingly excluded a role for ATG5 in ASM (based on the single western blot in Fig 1b) as stated in the Discussion on p. 13.

Response: We thank the Reviewer for this very insightful comment. Following the suggestion, we have now performed a new experiment demonstrating that knockout of SNAP23 in ATG7-KO cells reduced mitochondria secretion, thereby supporting our conclusion for the secretory autophagy route of mitochondria. Data from this new experiment are now presented in **Figure 5g** and we have revised the Results section accordingly (**Page 11, line 285-288**).

Regarding the role of ATG5 in ASM, please refer to our response to Q#10 by this Reviewer.

6. Figure 6: The western blots shown do not convincingly support the conclusions made. In panels a and b, inhibition of mitochondrial protein degradation in the DKO is variable (ie. protein dependent), appearing similar to the ATG7KO for some proteins. In particular, the comparison of panel d and e is concerning since the western blots appear highly similar yet the authors conclude that degradation was “almost completely blocked” (in d) versus “no effect on mitochondrial degradation” (in e).

Response: We thank the Reviewer for the critical comments. Following the advice, we have done the following:

First, we have repeated the experiments and added a WT control group for reference, with the new data presented in **Figure 6a**. The quantification data are provided in **Suppl Figure 5a**, which clearly demonstrate that the ATG14/ATG7 DKO cells display an inhibition of mitochondria degradation as compared to ATG7 KO cells.

Second, we have generated a new FIP200/ATG7 DKO cells (using the new dual sgRNA design as mentioned in response to Q#2 from this Reviewer) and repeated the experiments, with the new data presented in **Figure 6b**. The corresponding quantifications are presented in **Suppl Figure 5b**, which showed an inhibition of mitochondria degradation as compared to ATG7 KO cells.

Third, we have provided quantification for **Figures 6d** and **6e** in **Suppl Figure 5c** and **5d**, respectively, and the data show that knockout of ATG9A significantly inhibited mitochondria degradation, while RAB7A knockout had no significant effect on mitochondria degradation.

We have revised the Results section (**Page 11, line 303-310**) to reflect the above changes, as well as to revise the statement of “almost completely blocked” to “significantly inhibited” (**Page 11, line 307-308**).

7. Figure 7a: Quantitation, replicates and statistical analyses are required to back up the statement of “significantly higher levels of cGAS-STING activation”. While STING-dependent genes appear to be altered, these effects are not necessarily attributable to released mitochondrial components (mitochondrial DNA) as suggested.

Response: We thank the Reviewer for the suggestion. We have now provided quantifications for **Figure 7a** in **Figure 7b**.

To address the point whether the observed changes of the cGAS-STING pathway is directly related to the release of mitochondrial DNA, we have attempted multiple times to deplete mtDNA from our cell lines using existing protocols (Warren et al., 2017). Unfortunately, our cell lines were unable to survive this mtDNA depletion process, and we believe this was due to Parkin mitochondrial recruitment and subsequent Parkin-mediated cell death as previously described (Akabane et al., 2016; Carroll et al., 2014; Suen et al., 2010). We now have mentioned this limitation in the Discussion section (Page 14, line 403-405).

On the other hand, following the suggestion by Reviewer #3, we have generated new PINK1/ATG7 DKO cells, conducted new EV-transfer experiments and then measured the expression of STING-dependent genes and IL6 secretion in response to the addition of EVs from PINK1/ATG7 DKO cells. The new are presented in **Figure 7g and 7h**. The new results strengthen our conclusion given PINK1's specific role in targeting depolarized mitochondria. The Results section of the MS has been revised accordingly (Page 12, line 334-335).

8. Supplementary Fig4: WIPI2 puncta require quantitation to support the conclusion made in the text that they were "greatly reduced".

Response: We thank the Reviewer for the suggestion. We have now provided WIPI2 puncta quantifications in **Suppl Figure 4b**.

9. The isolated vesicles were not purified nor characterized sufficiently to be called exosomes. Throughout the text, extracellular or secreted "particles" (or small particles in the case of filtration) would be a better term since secreted material was not characterized as vesicular and likely contains many different components. This would also help to avoid confusion and improve consistency with the working model in Figure 8.

Response: We thank the Reviewer for the suggestion. While we agree that the term "extracellular particles" would be a better option based on the recommendations of MISEV2018, we worry that this may confuse the readers as this term has been used to specifically describe non-vesicular extracellular components (Choi et al., 2019; Hoshino et al., 2020; Malkin and Bratman, 2020). Additionally, the complete separation of EVs from other components by common EV isolation techniques such as ultracentrifugation remains a major limitation in the field as highlighted in MISEV2018. As such, we have decided to use the term EV (extracellular vesicle) fraction to label our preparations, although we acknowledge that our EV isolation method (ultracentrifugation followed by a wash in EV free buffer) likely contains some non-vesicular entities too.

Additionally, we have performed a new bottom-up iodixanol density gradient experiment to separate different EV populations and our new data showed that small EVs (CD9) and secreted mitochondria (ACO2 and COXII) indeed "floated" at different densities, suggesting that they are lipid-encapsulated structures. The new data have been presented in **Figure 4f**. We have revised the Results section accordingly (Page 9-10, line 251-255).

10. The authors refer to “mATG8 lipidation independence” throughout the text, but the majority of experiments were performed in ATG7KO (or DKO) backgrounds only, and assessed LC3B lipidation only. These statements should be corrected to “ATG7 independence” unless supported by appropriate additional lines of evidence.

Response: We thank the Reviewer for the critical comment.

Firstly, we have conducted new experiments and confirmed that the lipidation of the mATG8 orthologs LC3A, GABARAP and GABARAPL1 are also blocked in ATG7-KO, ATG5-KO and ATG3-KO cells. The new immunoblots are now provided in **Suppl Figure 1a and 1b**, and we have revised the Results section accordingly (**Page 6, line 139-140**).

Secondly, we have performed a new experiment to show that increased mitochondrial secretion occurs in ATG5-KO and ATG3-KO cells. This new data is now presented in **Suppl Figure 3a**, and we have revised the Results section accordingly (**Page 8, line 215-217**).

Lastly, we have corrected the phrase “mATG8 lipidation independence” to “ATG7 independence” where ATG7-KO cells were exclusively used in the manuscript accordingly (**Page 11, line 298; Page 14, line 404**).

11. Methods indicate that cells were treated with 100uM of CQ. This concentration seems very high as levels used are typically up to 10uM.

Response: We thank the Reviewer for the question. Here we would like to argue that the dosage of CQ used in our study is justifiable, based on the following:

Firstly, Ni and colleagues have previously reported that 10uM CQ was a non-saturating dose that did not completely suppress lysosomal activity, while higher doses of CQ (minimum of 50uM) was required to fully suppress lysosomal function (Ni et al., 2011).

Secondly, the dosage of 100uM CQ in autophagy studies is commonly used in the literature (Liang et al., 2014; Mauthe et al., 2018; Myeku and Figueiredo-Pereira, 2011). Therefore, we decided to keep this dosage of CQ unchanged.

Reviewer #3 (Remarks to the Author):

In this manuscript Tan et al. employ an impressive number of cell biological, biochemical and proteomics approaches as well as animal models to report the surprising finding that the clearance of damaged mitochondria occurs via secretion in cells lacking the ability to lipidate ATG8 family proteins. One of the main pathways for the removal of damaged mitochondria is termed PINK1/Parkin mediated mitophagy. In this pathway the damaged mitochondria are engulfed by autophagosomes. Canonically, this pathway entails the attachment of the ubiquitin-like ATG8 family proteins to membrane lipids residing in the nascent autophagosomal membrane. Here the authors delete key components of the ATG8 lipidation machinery (ATG7, ATG3, ATG5) and observe that damaged mitochondria can still be engulfed by autophagosomes, as observed before (PMID: 27864321). In addition, they make the unexpected observation that rather than accumulating autophagosomes containing mitochondria, these cells secrete them into the extracellular space in pathway the authors call ASM. A shortcoming of the study is that it remains unclear if ASM occurs also when mATG8 are present and can be lipidated? Is this an artificial pathway only observed in KO cells or a pathway that always occurs at low levels and is upregulated when the mATG8s can't be conjugated? Some evidence for the latter would make this very comprehensive study of interest for a much wider research community.

Response: First of all, we thank the Reviewer for the positive feedback and critical comments.

Regarding the point highlighted by the Reviewer on whether ASM also occurs in WT cells, we acknowledge that in this study we mainly focused ASM in mATG8-lipidation defective cells under acute mitochondrial damage. Nevertheless, our data also indicates the presence of some mitochondrial secretion from WT cells, at a much lower level comparing to mATG8-lipidation defective cells, upon acute mitochondrial damage. Since the amount of secreted mitochondria under such conditions is generally minimal, studying ASM in WT cells is technically difficult. Therefore, we believe that this topic could be further studied in the future when techniques of greater detection sensitivity are available.

Major comments:

1. To be absolutely sure that mATG8 lipidation is abolished in the ATG7, ATG3 and ATG5 KO cells the authors should blot for more mATG8 family members apart from LC3B, in particular as a faint band that runs close to the lipidated form of LC3B can be observed for these KO cell lines in Figure S1.

Response: We thank the Reviewer for the comment. We have conducted new experiments and our new data confirmed that the lipidation of the mATG8 orthologs LC3A, GABARAP and GABARAPL1 are also blocked in ATG7-KO, ATG5-KO and ATG3-KO cells. The new immunoblots are now provided in **Suppl Figure 1a** and **1b**. The Results section of the MS has been revised accordingly (**Page 6, line 139-140**).

2. Figure 2b: The TBK1 KO does not seem to be complete. How was the KO of TBK1 verified?

How can the authors therefore be sure that TBK1 is not required for mitochondrial clearance? Same for ATG9A (Figure 2c), which also doesn't seem to be completely depleted.

Response: We thank the Reviewer for the critical comment. To address this point, we have generated new TBK1 knockout cells and repeated the experiments for TBK1 and ATG9A-KO cells. The new data are presented in **Figure 2b** and **2c**, respectively. Quantifications of these results are presented in **Suppl Figure 2c and 2d**, respectively. The new data clearly demonstrate that TBK1 is not required for mitochondria clearance, while the knockout of ATG9A significantly inhibited mitochondria clearance.

3. There seems to be significant loss of mitochondrial IMM proteins in FIP200 KO cells. In order to fully appreciate the relative importance of the various autophagy factors the authors have analyzed regarding the clearance of mitochondria it would be important to quantify the blots.

Response: We thank the Reviewer for the comment. Similar point has also been raised by Reviewer #2. To address this point, we have generated new FIP200 KO cells by using a dual sgRNA design as previously described (Vargas et al., 2019), and presented the new data in **Figure 2d**. Additionally, we have provided quantifications for **Figure 2a to 2d** in **Suppl Figure 2b to 2e**, respectively, to support our conclusions as suggested.

4. Figure 2f: the authors should provide evidence that their BafA1 treatment effectively inhibits lysosomal degradation, for example by assessing p62 accumulation. In addition, can the authors please elaborate on why inhibition of proteasomal activity blocks clearance of all mitochondrial proteins.

Response: We thank the Reviewer for the comments and suggestions. Here we would like to address the points as follows:

First, we have now examined p62 levels and the new data are presented in **Figure 2f**.

Second, regarding the question on the role of the proteasome in mitochondria clearance, it has been well established that the ubiquitin proteasome system is required for Parkin-mediated mitophagy (Chan et al., 2011; McLelland et al., 2018; Rakovic et al., 2019). Mechanistically, this has been attributed to p97 extraction and subsequent proteasomal degradation of the OMM protein MFN2, thereby reducing mito-ER contacts to allow for increased accessibility of the PINK1/Parkin system to its substrates (McLelland et al., 2018). We have now elaborated this point in our Results section accordingly (**Page 7, line 179-181**).

5. Figs 3 and 4: The authors should determine the cell viability of the ATG7KO vs WT cells after A/O treatment to rule out that the increase of extracellular proteins including mitochondrial proteins is due to cell death associated cell lysis.

Response: We thank the Reviewer for the suggestion. To address the question of cell viability, we have performed new experiments to measure LDH release and found no difference among WT, ATG7-KO, ATG5-KO or ATG3-KO for cell viability within the control or

A/O-treated groups, suggesting that the increase of extracellular proteins is not due to cell lysis. The data is now presented in **Suppl Figure 3b**. We have revised the Results and Methods section accordingly (**Page 8, line 217-219**).

6. Fig S3a, Page 8 bottom: the authors write that they observe “single-membrane vesicular structures that resembled mitochondria with ruptured outer membranes”. This statement is confusing to this reviewer. If the mitochondria are released by secreted autophagosome then the outer autophagosomal membrane fuses with the plasma membrane leaving behind a single membrane vesicle (the former inner autophagosomal membrane) containing the damaged mitochondrion. Thus, in principle three membranes should be observed. Is this what the authors see? Unfortunately, the data shown in Fig S3 do not seem to support this.

Response: We thank the Reviewer for raising this insightful point. This is indeed an interesting and important question that has also baffled us. Interestingly, in early studies that characterized secretory autophagy, the cargo of interest (IL-1 β or ferritin) could be directly measured as a soluble factor by ELISA from the conditioned media, suggesting that those secretory autophagic cargos were not enveloped by a membrane (Dupont et al., 2011; Kimura et al., 2017). Similarly, the secretory autophagy cargo CTSD in glucocorticoid-induced cellular stress could also be measured directly by ELISA, again suggesting that CTSD was not enveloped by a vesicle (Martinelli et al., 2021).

In this study, our data tend to suggest that the secreted mitochondria were not engulfed by another layer of membrane such as the inner autophagosomal membrane as suggested by the Reviewer. How the secretory autophagosome fuses with the plasma membrane, or whether permeabilization of the inner autophagosome membrane occurs during secretory autophagy remains to be further studied in future.

We have revised the Discussion section of the MS to reflect this important point raised by the Reviewer (**Page 14, Line 375-382**).

7. Figure 6: the data should be quantified and directly compared. The block in IMM protein clearance by ATG14, FIP200 and ATG9A KO appears to be rather minor.

Response: We thank the Reviewer for the suggestion. Reviewer #2 had also raised a similar point. To address this, we have done the following:

First, we have repeated the experiments and added a WT control group for reference, with the new data presented in **Figure 6a**. The quantification data are provided in **Suppl Figure 5a**, which clearly demonstrate that the ATG14/ATG7 DKO cells display an inhibition of mitochondria degradation as compared to ATG7 KO cells or WT cells.

Second, we have generated a new FIP200/ATG7 DKO cells (using the new dual sgRNA design as mentioned in response to Q#3 from this Reviewer) and repeated the experiments, with the new data presented in **Figure 6b**. The corresponding quantifications are presented in **Suppl Figure 5b**, which showed an inhibition of mitochondria degradation as compared to ATG7 KO cells.

Third, we have provided quantification for **Figures 6d** and **6e** in **Suppl Figure 5c** and **5d**, respectively, and the data show that knockout of ATG9A significantly inhibited mitochondria degradation, while RAB7A knockout had no significant effect on mitochondria degradation.

We have revised the Results section (**Page 11, line 303-310**) to reflect the above changes, as well as to revise the statement of “almost completely blocked” to “significantly inhibited” (**Page 11, line 307-308**).

8. Figure 7: Did the authors correct for the higher number of EVs in ATG7 KO cells? Is it possible that the increased activation of the cGAS-STING pathway by the ATG7 KO derived EVs is due to a higher number of EVs? In this respect, it would be advisable to include an additional control cell line in which PINK1 is deleted on top of ATG7 in order to test if this activation is indeed due to secreted mitochondria.

Response: We thank the Reviewer for raising this point.

First, we acknowledge that in these experiments in Figure 7, we did not actually measure the number of EVs or perform normalization by EV numbers. Since we were investigating the effects of the gross EV fraction (which include secreted mitochondria) on recipient cells, we had performed normalization based on the cell number of EV-producing cells. We believed this way of normalization is more biologically relevant, as it measures the effect that the same number of cells from different genotypes have on another receiving population of cells.

Following the suggestion, we have generated PINK1/ATG7 DKO cells, conducted new EV-transfer experiments and then measured the expression of STING-dependent genes and IL6 secretion in response to the addition of EVs from PINK1/ATG7 DKO cells. As expected, no increase in STING-dependent genes and IL6 secretion was observed with EVs from PINK1/ATG7 DKO cells as compared to cells that received ATG7-KO EVs. The new data is now presented in **Figure 7g** and **7h**. We have revised our Results section accordingly (**Page 12, line 334-335**).

9. From Figure 3 on the authors employ solely ATG7 KO cells, which have defects in ATG8 lipidation but also fail to form the ATG12–ATG5 conjugate. This conjugate in turn may have functions that are not related to ATG8 lipidation. The authors should point out this complication more clearly in the manuscript.

Response: We thank the Reviewer for the comment. We have now included this complication in the Discussion section (**Page 14, line 389-390**).

Minor comments:

10. The authors should consider quantifying their key data derived from western blots.

Response: We thank the Reviewer for the suggestion. We have now provided immunoblot quantifications in **Figure 1b, 1d, 5b, 7b, Suppl Figure 2a to 2d** and **Suppl Figure 6a to 6d**.

11. Fig 3d: The legend should state more clearly what exactly is shown. Are these the GO analyses of the proteins enriched in the EVs of ATG7 KO vs WT cells?

Response: We thank the Reviewer for the clarification. Indeed, this is the GO analysis of the proteins enriched in the EVs of ATG7 KO vs WT cells. We have revised the figure legend of **Figure 3d** to more clearly describe this point (Page 30, line 939-940).

12. Page 10: The authors write that they used ATG9A as a cytosolic control. ATG9A is a transmembrane protein and as such unlikely to be cytosolic. In fact, the protein has been shown to localize to small Golgi derived vesicles. This sentence should be rephrased.

Response: We thank the Reviewer for the comment. We have now rephrased this statement to “ATG9A served as a control that is not incorporated into autophagosomes”

Reviewer #4 (Remarks to the Author):

Mitochondrial quality control (MQC) is key to the maintenance of mitochondrial homeostasis, and is critically involved in many pathophysiological processes. This manuscript identifies autophagic secretion of mitochondria (ASM) as a novel MQC mechanism through the clearance of damaged mitochondria in an ATG8 (LC3) lipidation-independent manner. The authors used quantitative proteomic analyses and profiled the protein content of the extracellular vesicles (EVs) derived from cells undergoing mitochondrial stress. In doing so, they were able to show that the damaged mitochondria were secreted via ASM, rather than the exosome-dependent pathway. Finally, they found that the ASM-derived mitochondria promoted innate immune signaling in a cGAS-STING-dependent manner. Although previous studies demonstrated that damaged mitochondria could be extruded from cells into extracellular vesicles, the current study is potentially interesting and provides a novel perspective and role of autophagic secretion in this process. Generally, the experiments shown are of good quality. The major weaknesses of the paper concern several mechanistic aspects of ASM and its physiological relevance. Certain experimental results were inconsistent with respect to the claims, and there were several key controls that were missing.

Response: We thank the Reviewer for the positive feedback and critical comments. We have revised our manuscript accordingly, as highlighted in details below.

Specific points are shown below:

1. The molecular mechanism of autophagosome-mitochondria fusion is unclear. When the lipidation of ATG8 (LC3) is blocked, the authors found that damaged mitochondria were sequestered by intact autophagosomes (or mitophagosomes), and were secreted to the extracellular space. However, previous studies have shown that the fusion of autophagosomes with mitochondria is mediated by lipidated LC3 with the adaptor proteins (e.g., P62, OPTN and NDP52) or mitochondrial proteins (e.g., BNIP3, NIX and FUNDC1) (Nature Cell Biology, volume 20, Pages 1013–1022 (2018)). The authors should explain which protein(s) might mediate this mitochondria-autophagosome fusion, besides ATG8 (LC3).

Response: We thank the Reviewer for the question.

First, we would like to argue that it might not be correct to define the engulfment of mitochondria into autophagosomes as “autophagosome-mitochondria fusion”.

Second, we would like to highlight that models of PINK1/Parkin mitophagy showing the recruitment of pre-formed autophagic membranes to mitochondria by interaction between mATG8s and the autophagy adaptor proteins NDP52 and OPTN have been updated with new discoveries. Fully intact autophagosomes that engulf mitochondria have been previously observed in the absence of mATG8s during PINK1-Parkin mitophagy (Nguyen et al., 2016). Additionally, it is now known that the autophagy adaptor proteins NDP52 and OPTN can directly recruit the ULK1 complex and ATG9A respectively to nucleate nascent autophagic membranes directly on targeted mitochondria (Vargas et al., 2019; Yamano et

al., 2020). We have revised our Discussion section accordingly to explain this (Page 13-14, line 372-375).

2. The biological relevance of ASM is not fully understood. In the EE model, which tissues/organs represent the source of the extracellular mitochondria? Related to the findings that ASM-derived mitochondria activate cGAS-STING signaling, did the authors observe any signs of inflammation in the EE mice? The authors should also add additional experiments demonstrating the physiological relevance of ASM, e.g., using the cardiomyocyte hypoxia model (<https://doi.org/10.1016/j.cmet.2021.08.002>).

Response: We thank the Reviewer for the critical comments. The source of extracellular mitochondria in our EE models is indeed a tough question to answer that would require extensive additional in-vivo work. As we have generated a muscle-specific ATG7 knockout mouse model, we can only speculate that circulating mitochondria originates from muscle, although we could not exclude the contribution from other organs/tissues at this point.

To partially address this point, we have conducted new experiments to measure serum IL6 levels in WT and ATG7-mKO mice both at resting (sedentary; SED) and with exhaustive exercise (EE). Although EE can marginally enhance IL6 level, there was no difference between the WT and ATG7-mKO mice. Here we provided this set data for the Reviewer's reference as **Response-Figure 1**.

Response-Figure 1. Control or muscle specific ATG7-KO (*Atg7f/f; Ckmm-cre*) mice were subjected to three consecutive days of exhaustive exercise. Serum IL6 was measured by the IL-6 Mouse ProQuantum Immunoassay Kit. Mean of $n = 5$ mice \pm SD is shown.

3. Some key controls were missing. As an example, for the SILAC experiment (Fig. 3A), the authors should add the wild type and ATG7 KO cells (without the A/O treatment) as the controls. Because only low amounts of extracellular vesicles were derived from these samples, the authors should explain how they performed the normalization procedures (at the sample preparation step and during bioinformatic analyses). The authors should also explain how they removed the secreted proteins (i.e., those derived from the conditional media) from the isolated extracellular vesicles. Finally, the authors should describe how they differentiated genuine secreted proteins from those that were non-specifically released from dead cells.

Response: We thank the Reviewer for critical comment.

For the SILAC experiment, we agree with the Reviewer that it would be better to include WT and ATG7KO cells without A/O treatment as controls. However, we decided not to work on cells without A/O treatment based on the following reasons: (i) The amount of secreted mitochondria in EVs is generally too low for this analysis and would not produce any meaningful data. (ii) SILAC is a quantitative proteomics technique that only allows for pairwise comparisons between a heavy and light isotope labelled cell population. Since our main aim was to compare the difference between the WT and ATG7 KO cells under A/O treatment which activates the PINK1-Parkin pathway, treated WT vs treated ATG7KO cells was the most relevant pairwise comparison to this study. We believe that the lack of non-treated controls (no A/O treatment) should not affect the conclusion of this study.

Following the advice, we have provided a more detailed description as follows: WT and ATG7KO cells were seeded at equal numbers per 150mm dish and treated as described. After collection of conditioned media, cells were lysed and protein quantifications were quantified as a proxy of cell number. The conditioned media from both samples were then mixed before EV isolation to avoid any variation that might arise from sample handling. The ratio of protein concentration from light and heavy cell lysates (it was essentially 1:1) were used to normalize the volume of conditioned media used for mixing before EV isolation and subsequent LC-MS. We have now provided more details of this in our **Methods section** (Page 20, line 552-557).

Moreover, to remove secreted proteins from our EV preparations, the initial ultra-centrifuged pellet was washed with PBS and re-spun at the same speed. The use of this EV-free buffer wash and respin step is a common method to remove soluble protein contaminants from the EV preparations (Thery et al., 2006; Wang et al., 2021). Description of this procedure has been provided in **Methods section** (Page 18, line 504-505).

4. In various panels of Figure 5 and Figure 6, the authors should also include WT cells as the controls.

Response: We thank the Reviewer for the critical comments. Following the advice, we have conducted new experiments to include a WT control group and the new data are presented in **Figure 6a**. Additionally, we have provided quantifications of protein changes for **Figure 6a, 6b, 6d and 6e** in **Suppl Figure. 5a to 5d**, respectively.

5. The authors should stain the mtDNA in the EVs or track the secretion of mitochondria using mtDNA staining.

Response: We thank the Reviewer for the suggestion. Actually, we have tried very hard on this experiment but unfortunately failed. The main technical issue is that current live-cell DNA labelling dyes (SYBR gold, SYBR green, pico green) to track mtDNA secretion would label both nuclear and mitochondria DNA. As such, it would be almost impossible to discern if the DNA found in EVs were indeed mtDNA during real time tracking of secretion.

6. The authors should measure the cell viability and membrane integrity of the A/O-treated

cells (in particular for the 24 hr-treatment group). This is an important consideration because there is a possibility that mtDNA was released, owing to cell death, rather than through the EVs.

Response: We thank the Reviewer for the suggestion. In fact, Reviewer #3 also raised a similar point. To address the question of cell viability, we have performed new experiments to measure LDH release and found no difference among WT, ATG7-KO, ATG5-KO or ATG3-KO for cell viability within the control or A/O-treated groups, suggesting that the increase of mitochondrial secretion is not due to cell lysis. The data is now presented in **Suppl Figure 3b**. We have revised the Results and Methods section accordingly (Page 8, line 217-219).

7. Page 11. The authors should elaborate the mechanism by which ASM induces the activation of cGAS-STING signaling in the recipient cells. The mtDNA is encapsulated in the EVs, and is not in direct contact with cGAS in the recipient cells. This is a potential gap in the model and the authors need to provide more evidence to substantiate their hypotheses.

Response: We thank the Reviewer for raising this point. While the activation of cGAS-STING by mtDNA-containing EVs have been reported previously (Rabas et al., 2021; Torralba et al., 2018), how EV-encapsulated mtDNA contacts cytosolic DNA sensor remains a topological mystery till date. We have now included this limitation of our study in the Discussion section accordingly (Page 14-15, line 405-408).

Other comments:

1. In several of the Figures (e.g., Fig 1A, 1B and 2A), there were two bands for mitoGFP (in particular in cells treated with A/O). The authors should explain this.

Response: We thank the Reviewer for the question. The release of free GFP fragments upon lysosome degradation (akin to the release of free GFP from GFP-LC3 during autophagy) results in the appearance of the lower band. We have now provided this description in the Results section (Page 5-6, line 129-132).

2. Fig. 1C, author should perform the LC3 staining and lysosome (LAMP1 or LAMP2) staining to check the co-localization of mtDNA with the autophagosomes and lysosomes.

Response: We thank the Reviewer for the suggestion. We have performed new immunostaining experiments to label HSP60 and LAMP2 in both WT and ATG7-KO cells, and the new data are presented in **Figure 2g and h**. We found that A/O treatment results in the engulfment of HSP60 puncta by LAMP2 rings in WT cells but not in ATG7-KO cells. This suggests that mitochondria are not sent for lysosomal degradation when ATG7 is absent, which is in-line with our model. We have revised our Results section accordingly (Page 7-8, line 189-194).

3. In several of the Figures (e.g., Fig S1A and S1B), the LC3I/LC3II levels were not changed upon the treatment of A/O. These results are inconsistent with the previous report (<https://doi.org/10.1016/j.cell.2016.11.042>). Furthermore, it seems that the LC3II level was very low in the control conditions. The authors need to discuss these results.

Response: We thank the Reviewer for the question.

Regarding the point on LC3II/LC3I levels, here we would like to argue that the LC3II/LC3I ratio is not recommended for measuring autophagy, based on the fact that the LC3I protein is not stable (Klionsky et al., 2021). In our study, the main purpose of measuring LC3B and other isoforms is to show the difference between the WT and ATG7/5/3 KO cells, rather than to show the changes of LC3II in response to A/O treatment.

In order to answer the question by the Reviewer, we performed a new time-course experiment to examine the changes of LC3II within a relative short period of time (up to 8 hrs) in WT cells and presented the data below (**Response-Figure 2**) for the Reviewer's reference. In this experiment, we have found that LC3-II levels were rapidly increased at 1hr and subsequent decreased back down to basal levels. In addition, as we used different concentrations of A/O and timepoints in comparison to the earlier report mentioned by the Reviewer (Wei et al., 2017), we believe that a direct comparison cannot be made given the dynamic flux of LC3 lipidation.

Response-Figure 2. HeLa cells stably expressing YFP-Parkin were treated with AO (1uM of antimycin and 1uM of oligomycin) for the indicated time points and immunoblotted for LC3B and B-actin.

4. Fig. 2C, It seems that ATG9A shows up as a smear. The authors should clearly label which corresponds to the ATG9A band.

Response: We thank the Reviewer for the suggestion. We have now labelled the ATG9A band.

5. The authors claimed that “The degradation of mitochondria markers was completely blocked in ATG9A-KO cells as compared to WT or ATG7-KO cells (Fig. 2c). Additionally, the knockout of FIP200 in cells impaired the degradation of mitochondrial proteins (Fig. 2d). However, in Fig. 2C and 2D, COXII and UQCRC2 were degraded even in the ATG9A KO or FIP200KO cells upon the 24-hr treatment of A/O. These results were inconsistent with the authors’ conclusion.

Response: We thank the Reviewer for the critical comment. In fact, Reviewer #2 also raised a similar point. To address this question, we have now provided immunoblot quantifications for **Figure 2c and 2d** in **Suppl Figure 2d and 2e**, respectively, to support our conclusion. We have also changed the term “completely blocked” to “significantly inhibited” for **Figure 2c** to tone down the claim. We have revised the Results section accordingly (**Page 7, line 162**).

Moreover, as we are unsure why our previous FIP200 knockouts showed a significant degradation of CS specifically, we have generated new FIP200 KO cells by using a dual sgRNA design as previously described (Vargas et al., 2019) and performed new experiments with this new FIP200 KO cells. The new data are now presented in **Figure 2d** with the corresponding quantification data in **Suppl Figure 2e**.

6. The authors claimed that “Inhibition of lysosomal activity with lysosome inhibitors bafilomycin A1 (BafA1) or chloroquine (CQ) resulted in the partial impairment of IMM and matrix protein degradation induced by A/O, while the degradation of OMM proteins were unaffected.” However, in Fig. 2F and Fig. S2d, the degradation of CS, COXII and UQCRC2 was partly blocked by BafA1 in ATG7KO and ATG5/ATG3-KO cells. The authors need to explain these observations.

Response: We thank the Reviewer for raising this important. To address this issue, we have now provided quantifications for immunoblots of CS, UQCRC2 and COXII for **Figure 2f** in **Suppl Figure 2f**. Our data show that BafA1 could partially blocked degradation of IMM and matrix proteins in WT cells, but not in ATG7-KO cells, which is consistent with our description of the data.

7. Fig. 3c and 3d, many of the identified proteins were derived from the lysosomes (e.g., mTOR, LAMTOR1, LAMP1 and LAMP2). The authors should address the possibility that the damaged mitochondria might fuse with the autophagosome-lysosomes?

Response: We thank the Reviewer for the suggestion. We have addressed this question in Q#2 of the “other comments” section above.

8. Fig. 3, the authors did not identify ACO2, COXII, Tom70, Tom20, MFN1, CD9 or CD63 in the MS-based proteomic experiments. These results were inconsistent with the authors’ conclusion that these proteins were increased in the EVs from ATG7 KO cells.

Response: We thank the Reviewer for raising this important question. We acknowledge that those mitochondrial proteins were not picked up by the MS-based proteomic experiments. There are several possible reasons to explain:

First, it is probably due to the technical limitation of this method. As we all know, the dynamic range of protein abundance in the cell can span up to seven orders of magnitude. Given that MS instruments are unable to cover the full dynamic range of the proteome (they generally cover four orders of magnitude of the most abundant end of proteins), exponentially increasing amounts of starting material are required to reach full-depth coverage. This known drawback has been eloquently described by Zubarev et al (Zubarev, 2013).

Second, this technical challenge is further compounded by the issue of low protein yields in EV isolates despite using large amounts of cells. Third and maybe more importantly, many of the proteins listed by the Reviewer are actually OMM proteins (Tom70, Tom20, MFN1) and our data in Figure 1 clearly show that the proteasomal degradation of OMM proteins are not affected by deletion of key lipidation ATGs, and thus would not be detected by MS.

Therefore, we believe that the lack of detection of certain proteins in MS-based experiments might not reflect the biological truth. Rather, the overall proteomic landscape of the EV population should be looked at in an untargeted way (as we have done) to reach a conclusion, given that we are not investigating the secretion of a specific protein.

9. Fig. 4A, the level of CD63 was lower in both WCL and EVs from the ATG7 KO cells without A/O treatment, compared to the WT cells. The authors should repeat these experiments and check the expression of CD63.

Response: We thank the Reviewer for the suggestion. We have now re-run the immunoblots for CD63 and show that CD63 expression in the WCL of WT and ATG7-KO cells are similar. The new data are presented in **Figure 4a** accordingly.

10. Fig. 4b, the level of LC3 was not changed upon EE in the WT mice. The authors need to explain this observation.

Response: We thank the Reviewer for highlighting this point. The response of LC3 lipidation to exercise in the literature has been diverse. While some studies have shown an increase in LC3 lipidation after exercise (He et al., 2012; Pagano et al., 2014), several other studies have demonstrated that LC3-II does not increase immediately after exercise in various rodent models and humans (Brandt et al., 2018; Kim et al., 2012; Zhang et al., 2019). This suggests that changes in LC3 lipidation may depend on the exercise protocol as well as time of assessment after exercise. As mentioned above in Q#3, the main purpose of assessing LC3B was to confirm the difference in lipidation status between WT and ATG7 KO genotypes.

11. Fig. 5A, the authors should explain why there was less protection of NDP52 against proteinase K digestion in ATG7KO cells compared to that in WT cells. Based on the hypothesis proposed by the authors, NDP52 should be equally protected against proteinase K digestion under these two conditions.

Response: We thank the Reviewer for raising this important point. We believe that our data (Figure 5a) showing less protection of NDP52 against proteinase K digestion in ATG7KO cells compared to that in WT cells are in fact consistent with the literature. It has been previously reported that cells devoid of all six mATG8s had less protection of NDP52 than WT cells. This was attributed to a lower autophagosomal volume and therefore lower cargo capacity. (Nguyen et al., 2016). We have added this reference as a possible explanation for the reduced protection of NDP52 we observed in ATG7-KO cells, and have revised the Results section accordingly (**Page 10, line 268-269**).

12. The pattern of ACO2 in Figure 5d is different from that in the other Figures (e.g., Figure 5C, 5E and 5F). The authors should repeat these western blotting experiments.

Response: We thank the reviewer for the comment. We have now re-run the immunoblot experiment and present a cleaner ACO2 immunoblot in **Figure 5d**.

13. Fig. 6, the authors should add the WT cells as the control. The authors claimed that

“When compared to ATG7-KO cells, the degradation of IMM and matrix proteins were significantly inhibited in ATG7/ATG14-DKO or ATG7/FIP200-DKO cells upon A/O treatments (Fig. 6a-6b). However, it seems that the degradation of matrix and IMM proteins were not dramatically affected in the ATG7/FIP200-DKO cells upon 24-hrs of A/O treatment. The authors should clarify this.

Response: We thank the Reviewer for the critical comments. Following the suggestion, we have repeated the experiments with addition of a WT group for reference, and the new data are presented in **Figure 6a**. Moreover, we have generated new ATG7/FIP200-DKO cells and repeated the experiments, and the new data are presented in **Figure 6b**. We have also provided the protein quantifications for Figure 6a and 6b in **Suppl Figure 5a and 5b**, respectively.

14. Fig. 7, the authors should measure and stain the mtDNA in the EVs. The authors need to demonstrate how mtDNA is released from the EVs to activate cGAS-STING in the recipient cells.

Response: We have addressed these points regarding the measurement of mtDNA in EVs in our Response to Q#5 and Q#7 of the “Specific points” section from this Reviewer above.

15. Fig.7C, the authors should exclude the possibility that a portion of IL6 is derived from the EVs during the isolation step.

Response: We thank the Reviewer for raising this technical question. Following the suggestion, we have now added an additional “no cell control group” where isolated EVs were added to wells containing cell culture media with no recipient cells. We show that negligible levels of IL6 were detected in this control group, confirming that IL6 was indeed produced by recipient cells and not derived from EVs. The new data is presented in **Figure 7h**. We have revised the Results section accordingly (**Page 12, line 334-335**).

References

- Akabane, S., Matsuzaki, K., Yamashita, S., Arai, K., Okatsu, K., Kanki, T., Matsuda, N., and Oka, T. (2016). Constitutive Activation of PINK1 Protein Leads to Proteasome-mediated and Non-apoptotic Cell Death Independently of Mitochondrial Autophagy. *J Biol Chem* *291*, 16162-16174.
- Bonsergent, E., and Lavieu, G. (2019). Content release of extracellular vesicles in a cell-free extract. *FEBS Lett* *593*, 1983-1992.
- Brandt, N., Gunnarsson, T.P., Bangsbo, J., and Pilegaard, H. (2018). Exercise and exercise training-induced increase in autophagy markers in human skeletal muscle. *Physiol Rep* *6*, e13651.
- Carroll, R.G., Hollville, E., and Martin, S.J. (2014). Parkin sensitizes toward apoptosis induced by mitochondrial depolarization through promoting degradation of Mcl-1. *Cell Rep* *9*, 1538-1553.
- Chan, N.C., Salazar, A.M., Pham, A.H., Sweredoski, M.J., Kolawa, N.J., Graham, R.L., Hess, S., and Chan, D.C. (2011). Broad activation of the ubiquitin-proteasome system by Parkin is critical for mitophagy. *Hum Mol Genet* *20*, 1726-1737.
- Choi, D., Montermini, L., Jeong, H., Sharma, S., Meehan, B., and Rak, J. (2019). Mapping Subpopulations of Cancer Cell-Derived Extracellular Vesicles and Particles by Nano-Flow Cytometry. *ACS Nano* *13*, 10499-10511.
- Dupont, N., Jiang, S., Pilli, M., Ornatowski, W., Bhattacharya, D., and Deretic, V. (2011). Autophagy-based unconventional secretory pathway for extracellular delivery of IL-1beta. *EMBO J* *30*, 4701-4711.
- Furman, D., Campisi, J., Verdin, E., Carrera-Bastos, P., Targ, S., Franceschi, C., Ferrucci, L., Gilroy, D.W., Fasano, A., Miller, G.W., *et al.* (2019). Chronic inflammation in the etiology of disease across the life span. *Nat Med* *25*, 1822-1832.
- He, C., Bassik, M.C., Moresi, V., Sun, K., Wei, Y., Zou, Z., An, Z., Loh, J., Fisher, J., Sun, Q., *et al.* (2012). Exercise-induced BCL2-regulated autophagy is required for muscle glucose homeostasis. *Nature* *481*, 511-515.
- Hoshino, A., Kim, H.S., Bojmar, L., Gyan, K.E., Cioffi, M., Hernandez, J., Zambirinis, C.P., Rodrigues, G., Molina, H., Heissel, S., *et al.* (2020). Extracellular Vesicle and Particle Biomarkers Define Multiple Human Cancers. *Cell* *182*, 1044-1061 e1018.
- Im, H., Shao, H., Park, Y.I., Peterson, V.M., Castro, C.M., Weissleder, R., and Lee, H. (2014). Label-free detection and molecular profiling of exosomes with a nano-plasmonic sensor. *Nat Biotechnol* *32*, 490-495.
- Joshi, A.U., Minhas, P.S., Liddelow, S.A., Haileselassie, B., Andreasson, K.I., Dorn, G.W., 2nd, and Mochly-Rosen, D. (2019). Fragmented mitochondria released from microglia trigger A1 astrocytic response and propagate inflammatory neurodegeneration. *Nat Neurosci* *22*, 1635-1648.
- Kim, Y.A., Kim, Y.S., and Song, W. (2012). Autophagic response to a single bout of moderate exercise in murine skeletal muscle. *J Physiol Biochem* *68*, 229-235.
- Kimura, T., Jia, J., Kumar, S., Choi, S.W., Gu, Y., Mudd, M., Dupont, N., Jiang, S., Peters, R., Farzam, F., *et al.* (2017). Dedicated SNAREs and specialized TRIM cargo receptors mediate secretory autophagy. *EMBO J* *36*, 42-60.
- Klionsky, D.J., Abdel-Aziz, A.K., Abdelfatah, S., Abdellatif, M., Abdoli, A., Abel, S., Abeliovich, H., Abildgaard, M.H., Abudu, Y.P., Acevedo-Arozena, A., *et al.* (2021). Guidelines for the use and interpretation of assays for monitoring autophagy (4th edition)(1). *Autophagy* *17*, 1-382.
- Kowal, J., Arras, G., Colombo, M., Jouve, M., Morath, J.P., Prindal-Bengtson, B., Dingli, F., Loew, D., Tkach, M., and Thery, C. (2016). Proteomic comparison defines novel markers to characterize heterogeneous populations of extracellular vesicle subtypes. *Proc Natl Acad Sci U S A* *113*, E968-977.
- Kwon, H.S., and Koh, S.H. (2020). Neuroinflammation in neurodegenerative disorders: the roles of microglia and astrocytes. *Transl Neurodegener* *9*, 42.
- Levine, B., and Kroemer, G. (2019). Biological Functions of Autophagy Genes: A Disease Perspective. *Cell* *176*, 11-42.
- Liang, X., Tang, J., Liang, Y., Jin, R., and Cai, X. (2014). Suppression of autophagy by chloroquine sensitizes 5-fluorouracil-mediated cell death in gallbladder carcinoma cells. *Cell Biosci* *4*, 10.
- Malkin, E.Z., and Bratman, S.V. (2020). Bioactive DNA from extracellular vesicles and particles. *Cell Death Dis* *11*, 584.
- Martinelli, S., Anderzhanova, E.A., Bajaj, T., Wiechmann, S., Dethloff, F., Weckmann, K., Heinz, D.E., Ebert, T., Hartmann, J., Geiger, T.M., *et al.* (2021). Stress-primed secretory autophagy promotes extracellular BDNF maturation by enhancing MMP9 secretion. *Nat Commun* *12*, 4643.
- Mathieu, M., Nevo, N., Jouve, M., Valenzuela, J.I., Maurin, M., Verweij, F.J., Palmulli, R., Lankar, D., Dingli, F., Loew, D., *et al.* (2021). Specificities of exosome versus small ectosome secretion revealed by live intracellular tracking of CD63 and CD9. *Nat Commun* *12*, 4389.

- Mauthe, M., Orhon, I., Rocchi, C., Zhou, X., Luhr, M., Hijlkema, K.J., Coppes, R.P., Engedal, N., Mari, M., and Reggiori, F. (2018). Chloroquine inhibits autophagic flux by decreasing autophagosome-lysosome fusion. *Autophagy* *14*, 1435-1455.
- McLelland, G.L., Goiran, T., Yi, W., Dorval, G., Chen, C.X., Lauinger, N.D., Krahn, A.I., Valimehr, S., Rakovic, A., Rouiller, I., *et al.* (2018). Mfn2 ubiquitination by PINK1/parkin gates the p97-dependent release of ER from mitochondria to drive mitophagy. *Elife* *7*.
- Myeku, N., and Figueiredo-Pereira, M.E. (2011). Dynamics of the degradation of ubiquitinated proteins by proteasomes and autophagy: association with sequestosome 1/p62. *J Biol Chem* *286*, 22426-22440.
- Nguyen, T.N., Padman, B.S., Usher, J., Oorschot, V., Ramm, G., and Lazarou, M. (2016). Atg8 family LC3/GABARAP proteins are crucial for autophagosome-lysosome fusion but not autophagosome formation during PINK1/Parkin mitophagy and starvation. *J Cell Biol* *215*, 857-874.
- Ni, H.M., Bockus, A., Wozniak, A.L., Jones, K., Weinman, S., Yin, X.M., and Ding, W.X. (2011). Dissecting the dynamic turnover of GFP-LC3 in the autolysosome. *Autophagy* *7*, 188-204.
- Pagano, A.F., Py, G., Bernardi, H., Candau, R.B., and Sanchez, A.M. (2014). Autophagy and protein turnover signaling in slow-twitch muscle during exercise. *Med Sci Sports Exerc* *46*, 1314-1325.
- Peruzzotti-Jametti, L., Bernstock, J.D., Willis, C.M., Manferrari, G., Rogall, R., Fernandez-Vizarra, E., Williamson, J.C., Braga, A., van den Bosch, A., Leonardi, T., *et al.* (2021). Neural stem cells traffic functional mitochondria via extracellular vesicles. *PLoS Biol* *19*, e3001166.
- Rabas, N., Palmer, S., Mitchell, L., Ismail, S., Gohlke, A., Riley, J.S., Tait, S.W.G., Gammage, P., Soares, L.L., Macpherson, I.R., *et al.* (2021). PINK1 drives production of mtDNA-containing extracellular vesicles to promote invasiveness. *J Cell Biol* *220*.
- Rakovic, A., Ziegler, J., Martensson, C.U., Prasuhn, J., Shurkewitsch, K., Konig, P., Paulson, H.L., and Klein, C. (2019). PINK1-dependent mitophagy is driven by the UPS and can occur independently of LC3 conversion. *Cell Death Differ* *26*, 1428-1441.
- Suen, D.F., Narendra, D.P., Tanaka, A., Manfredi, G., and Youle, R.J. (2010). Parkin overexpression selects against a deleterious mtDNA mutation in heteroplasmic cybrid cells. *Proc Natl Acad Sci U S A* *107*, 11835-11840.
- Thery, C., Amigorena, S., Raposo, G., and Clayton, A. (2006). Isolation and characterization of exosomes from cell culture supernatants and biological fluids. *Curr Protoc Cell Biol* *Chapter 3*, Unit 3 22.
- Torralba, D., Baixauli, F., Villarroya-Beltri, C., Fernandez-Delgado, I., Latorre-Pellicer, A., Acin-Perez, R., Martin-Cofreces, N.B., Jaso-Tamame, A.L., Iborra, S., Jorge, I., *et al.* (2018). Priming of dendritic cells by DNA-containing extracellular vesicles from activated T cells through antigen-driven contacts. *Nat Commun* *9*, 2658.
- Vargas, J.N.S., Wang, C., Bunker, E., Hao, L., Maric, D., Schiavo, G., Randow, F., and Youle, R.J. (2019). Spatiotemporal Control of ULK1 Activation by NDP52 and TBK1 during Selective Autophagy. *Mol Cell* *74*, 347-362 e346.
- Wang, F., Cerione, R.A., and Antonyak, M.A. (2021). Isolation and characterization of extracellular vesicles produced by cell lines. *STAR Protoc* *2*, 100295.
- Warren, E.B., Aicher, A.E., Fessel, J.P., and Konradi, C. (2017). Mitochondrial DNA depletion by ethidium bromide decreases neuronal mitochondrial creatine kinase: Implications for striatal energy metabolism. *PLoS One* *12*, e0190456.
- Wei, Y., Chiang, W.C., Sumpter, R., Jr., Mishra, P., and Levine, B. (2017). Prohibitin 2 Is an Inner Mitochondrial Membrane Mitophagy Receptor. *Cell* *168*, 224-238 e210.
- Yamano, K., Kikuchi, R., Kojima, W., Hayashida, R., Koyano, F., Kawawaki, J., Shoda, T., Demizu, Y., Naito, M., Tanaka, K., *et al.* (2020). Critical role of mitochondrial ubiquitination and the OPTN-ATG9A axis in mitophagy. *J Cell Biol* *219*.
- Zhang, D., Lee, J.H., Kwak, S.E., Shin, H.E., Zhang, Y., Moon, H.Y., Shin, D.M., Seong, J.K., Tang, L., and Song, W. (2019). Effect of a Single Bout of Exercise on Autophagy Regulation in Skeletal Muscle of High-Fat High-Sucrose Diet-Fed Mice. *J Obes Metab Syndr* *28*, 175-185.
- Zubarev, R.A. (2013). The challenge of the proteome dynamic range and its implications for in-depth proteomics. *Proteomics* *13*, 723-726.

REVIEWERS' COMMENTS

Reviewer #1 (Remarks to the Author):

The authors have adequately addressed all my concerns by the addition of new data and controls

Reviewer #2 (Remarks to the Author):

The addition of quantitation, statistical analyses, new KO lines, new experiments and discussion (noting limitations) have greatly improved the manuscript. The authors have comprehensively and satisfactorily addressed my concerns.

Reviewer #3 (Remarks to the Author):

The authors have addressed the points I raised and I have no further comments.

Reviewer #4 (Remarks to the Author):

The authors have generated additional data to support their conclusions. In addition, the author also included further discussions to address and clarify the previous concerns. The paper has been greatly improved, and is ready to be published.

We would like to thank all reviewers for their time and suggestions, which has greatly improved our manuscript.

Reviewer #1 (Remarks to the Author):

The authors have adequately addressed all my concerns by the addition of new data and controls

We thank the reviewer for the suggestions on additional supporting experiments and positive comments.

Reviewer #2 (Remarks to the Author):

The addition of quantitation, statistical analyses, new KO lines, new experiments and discussion (noting limitations) have greatly improved the manuscript. The authors have comprehensively and satisfactorily addressed my concerns.

We thank the reviewer for the suggestions on additional supporting experiments and positive comments.

Reviewer #3 (Remarks to the Author):

The authors have addressed the points I raised and I have no further comments.

We thank the reviewer for the suggestions on additional supporting experiments and positive comments.

Reviewer #4 (Remarks to the Author):

The authors have generated additional data to support their conclusions. In addition, the author also included further discussions to address and clarify the previous concerns. The paper has been greatly improved, and is ready to be published.

We thank the reviewer for the suggestions on additional supporting experiments and positive comments.